

# Craniofacial ontogeny in Tylosaurinae

Amelia R. Zietlow

Department of Biology, Carthage College, Kenosha, WI, USA

## ABSTRACT

Mosasaurs were large, globally distributed aquatic lizards that lived during the Late Cretaceous. Despite numerous specimens of varying maturity, a detailed growth series has not been proposed for any mosasaur taxon. Two taxa—*Tylosaurus proriger* and *T. kansasensis/nepaeolicus*—have robust fossil records with specimens spanning a wide range of sizes and are thus ideal for studying mosasaur ontogeny. *Tylosaurus* is a genus of particularly large mosasaurs with long, edentulous anterior extensions of the premaxilla and dentary that lived in Europe and North America during the Late Cretaceous. An analysis of growth in *Tylosaurus* provides an opportunity to test hypotheses of the synonymy of *T. kansasensis* with *T. nepaeolicus*, sexual dimorphism, anagenesis, and heterochrony. Fifty-nine hypothetical growth characters were identified, including size-dependent, size-independent, and phylogenetic characters, and quantitative cladistic analysis was used to recover growth series for the two taxa. The results supported the synonymy of *T. kansasensis* with *T. nepaeolicus* and that *T. kansasensis* represent juveniles of *T. nepaeolicus*. A Spearman rank-order correlation test resulted in a significant correlation between two measures of size (total skull length and quadrate height) and maturity. Eleven growth changes were shared across both species, neither of the ontogram topologies showed evidence of skeletal sexual dimorphism, and a previous hypothesis of paedomorphy in *T. proriger* was not rejected. Finally, a novel hypothesis of anagenesis in Western Interior Seaway *Tylosaurus* species, driven by peramorphy, is proposed here.

Corresponding author
Amelia R. Zietlow,
azietlow@carthage.edu

## INTRODUCTION

### Mosasaur ontogeny

Mosasaurs (Squamata: Mosasauridae) were a group of large, predatory marine lizards with a global distribution that lived during the Late Cretaceous. The fossils of several taxa span a wide range of sizes and therefore are presumably of varying maturity. The first published study of growth in mosasaurs was done by *Caldwell (1996)*, which sought to determine the patterns of ossification in the autopodial skeleton across mosasauroids and to test the congruence between these growth processes and mosasaur phylogeny. *Caldwell (1996)* found that many ossified carpals and tarsals is the ancestral condition, whereas more derived species have less ossified carpals and tarsals; also, a low number of ossified carpals and tarsals is characteristic of juveniles.

*Pellegrini (2007)* published the first study of osteohistology in mosasaur limb bones. By counting lines of arrested growth in specimens of *Tylosaurus*, *Platecarpus*, and *Clidastes*, the author found that mosasaur growth was initially fast, and then slowed when the animals reached 5–7 years old; they also noted that the rate of growth is faster overall than extant terrestrial squamates. The decrease in growth rate is interpreted as the onset of sexual maturity, given that 5–7 years is also the onset of sexual maturity in large extant varanid lizards. However, no proxies for maturity beyond chronological age were explicitly given.

*Houssaye & Tafforeau (2012)* examined vertebral microanatomy to test the hypothesis that juvenile mosasaurs inhabited shallower environments than adults; in other marine reptiles, an ontogenetic shift from shallow habitats to deeper ones was inferred through progressive loss of bone mass (*Wiffen et al., 1995*). The authors acknowledged that the assessment of maturity is based on size alone, given that skeletochronology is not reliable in mosasaur vertebrae due to a high amount of inner bone resorption (*Houssaye & Tafforeau, 2012*). They found that vertebral microstructure is similar between juveniles and adults, which implied that juveniles were as agile swimmers as adults and, therefore, the authors rejected the hypothesis that juvenile mosasaurs were restricted to shallow, sheltered nurseries. They also noted that, relative to other squamates, mosasaur vertebrae seem to be paedomorphic in that there is a general inhibition of bone remodeling.

*Harrell & Martin (2015)* described a *Mosasaurus hoffmannii* specimen found in South Dakota, which significantly extended the geographic range of the taxon farther north in the Western Interior Seaway (WIS). In addition to a description of the skull, the authors identified several ontogenetically variable characters, including the shape of the frontal in dorsal view, dentary depth, and the shape of a notch on the anterolateral flange of the coronoid. The abstract mentions that the shape of the supratemporal fenestrae also varies with maturity, but this is not mentioned anywhere else in the article. The authors provided growth series that showed the growth changes associated with frontal shape and the anterolateral notch of the coronoid, but they are limited to three and four specimens, respectively. Although variation in the quadrate is noted, they did not consider it to be ontogenetic.

*Jiménez-Huidobro, Simões & Caldwell (2016)* proposed that specimens of two sympatric species of *Tylosaurus*, *T. kansasensis* (*Everhart, 2005*) and *T. nepaeolicus*, are synonymous, and that *T. kansasensis* specimens are juveniles. They identified several characters in *T. kansasensis* that purportedly show the juvenile conditions seen in another species of *Tylosaurus*, *T. proriger*, and concluded that there are "no differences between the two nominal species that cannot be attributed to size, and thus ontogenetic stage" (*Jiménez-Huidobro, Simões & Caldwell, 2016*: 80), and that *T. kansasensis* are therefore juveniles of *T. nepaeolicus*. Also, the authors suggested that *T. proriger* may be paedomorphic relative to *T. nepaeolicus* due to the presence of a dorsal midline crest on the frontal and convex lateral borders of the parietal table, features purportedly seen in *T. kansasensis*, but not *T. nepaeolicus*. The authors provided no justification (or references to one) for identifying one *T. proriger* specimen, RMM 5610, as a juvenile, and all others

(e.g., AMNH FARB 4909) as adults. The following characters were proposed to be ontogenetically variable: definition of the parietal nuchal fossa; medial curvature of the quadrate suprastapedial process; thickness of the quadrate suprastapedial process; thickness of the frontal posterolateral processes; shape of the lateral borders of the parietal table; and presence of the frontal dorsal midline crest. Despite identifying these characters, the authors do not propose a growth series of individual specimens.

*Carpenter (2017)* described the vertebral morphology of several specimens of *T. proriger*, including a purported juvenile, RMM 5610. The goal was to deduce the method of swimming of this species by analyzing the degree of vertebral mobility. In addition to providing evidence that adult *T. proriger* were carangiform swimmers (propulsion generated by movement of the hips and tail), differences were seen in the vertebral mobility of RMM 5610, suggesting a faster, tail-driven method of swimming in juveniles.

*Green (2018)* proposed a growth series of four specimens of *Clidastes* sp. that was based on histological data. Using cyclical growth marks, the author concluded that the growth rate in *Clidastes* was rapid during its first year of life, moderate between the second and sixth years, and slow from the seventh year onward. Based on growth rates, it was hypothesized that mosasaurs were ectothermic. These results are similar to those of *Pellegrini (2007)*, however, like the earlier study, they are limited by a small sample size (number of specimens) and no estimates of maturity beyond size and chronological age.

A description of the smallest known *Tylosaurus* specimen (FHSM VP-14845) was published by *Konishi, Jiménez-Huidobro & Caldwell (2018)*. Although it is not identifiable to species, it shares many features with *Tylosaurus* generally, especially with the juvenile *T. proriger* specimen, RMM 5610. The authors determined that the specimen is most likely a neonate (newborn) using an estimated total body length and neonate-to-maternal body length proportion data from extant varanid lizards. The authors rejected the possibility that the length of the premaxillary predental rostrum is sexually dimorphic due to its presence in this extremely young individual, but they did note that it is proportionally much shorter than what is seen in adult specimens. Also, they note that the second set of premaxillary teeth are posterolateral to the first set, rather than posterior to them, and that the shape of the premaxillary rostrum is a "gently pointed arc" in dorsal view (*Konishi, Jiménez-Huidobro & Caldwell, 2018*: 3).

*Stewart & Mallon (2018)* described two purported subadult specimens of *T. proriger* (CMN 8162 and CMN 51258-51263) and hypothesized the growth pattern of various skull structures. The study revealed a significant correlation of all individual bone measurements with total skull length (TSL), as well as isometric growth for all characters except quadrate height (QH), which was found to be positively allometric, and premaxillary predental rostrum length, which was found to be negatively allometric. They also rejected the hypothesis that *T. kansasensis* represent juveniles of *T. nepaeolicus* (*Jiménez-Huidobro, Simões & Caldwell, 2016*), stating that the growth trends between *T. kansasensis* and *T. nepaeolicus* do not match what is seen in *T. proriger*, and that there is not enough evidence to support the proposed ontogenetic characters.

## Assessment

Overall, there is a deficit of literature devoted to growth in any individual mosasaur taxon, and despite the several papers that do address growth in mosasaurs, the topic remains poorly understood. Little to no justification beyond size or histological data is given for determining the relative maturity of specimens, and growth stages are limited to the vague categories of "juvenile," "subadult," and "adult." No study thus far has attempted to combine all types of data—size, proportional, and size-independent (i.e., morphological)—using an objective, quantifiable, and replicable method to recover a growth series for any mosasaur species. In addition to enhancing our understanding of mosasaur ontogeny, such an analysis could prove particularly useful in resolving the validity of certain species (in this case, *T. kansasensis*) and the presence or absence of sexual dimorphism.

### Tylosaurus proriger

*Tylosaurus proriger* was a particularly large mosasaur—the largest individual, the "Bunker" specimen (KUVP 5033), has an estimated TSL of 1.7 m (Table 1)—that lived in the WIS during the upper Santonian to the middle Campanian, between 84 and 80 million years ago (Ma) (*Jiménez-Huidobro & Caldwell, 2019*). The type specimen of *T. proriger* (MCZ 4374) was described by *Cope (1869)* and includes a partial snout, cranial fragments, and thirteen vertebrae (*Russell, 1967*). Cope originally named the species *Macrosaurus proriger*. The genus was formally changed by *Leidy (1873)* to *Tylosaurus* ("knob lizard"), of which *T. proriger* is the type species (*Everhart, 2017*).

Tylosaurus proriger is an unquestionably valid taxon diagnosed by the following suite of cranial characters: (1) premaxilla-maxilla suture ends posterior to the fourth maxillary tooth; (2) quadrate suprastapedial process reaches half the length of the complete bone; (3) quadrate infrastapedial process is moderately developed; (4) quadrate tympanic ala is thin; (5) medial crest of the frontal is well-developed; (6) prefrontal overlaps the postorbitofrontal; (7) dorsal, medial, and lateral invasion of the parietal by frontal alae; and (8) teeth that lack flutes (*Russell, 1967*; *Jiménez-Huidobro & Caldwell, 2019*).

### Tylosaurus kansasensis and Tylosaurus nepaeolicus

*Tylosaurus kansasensis* and *T. nepaeolicus* are both known from the WIS during the upper Coniacian to the lower Santonian, from 88 to 85 Ma (*Everhart, 2017*; *Jiménez-Huidobro & Caldwell, 2019*). The type specimen of *T. nepaeolicus* (AMNH FARB 1565) was described by *Cope (1874)* and includes a quadrate, jaw fragments, rib fragment, and single dorsal vertebra (*Russell, 1967*; *Jiménez-Huidobro, Simões & Caldwell, 2016*). The type specimen of *T. kansasensis* (FHSM VP-2295) was described by *Everhart (2005)* and consists of an articulated skull and six associated cervical vertebrae.

Tylosaurus nepaeolicus is diagnosed by the following cranial characters: (1) premaxilla–maxilla suture ends posteriorly above midpoint between third and fourth maxillary teeth; (2) prefrontal overlaps the postorbitofrontal; (3) frontal with dorsal midline crest poorly developed or absent in adult; (4) lateral borders of parietal table slightly convex; (5) ectopterygoid does not contact the maxilla; (6) infrastapedial process of

Table 1 Measurements, in millimeters, of all specimens included in this project for which measurement data was available.

| Specimen | A | B | C | D | E | F | G | H |
|---|---|---|---|---|---|---|---|---|
| *Tylosaurus* sp. | | | | | | | | |
| FHSM VP-14845 | 300** | 3 | ? | 30* | ? | ? | ? | ? |
| FHSM VP-14841 | ? | 13 | ? | ? | ? | ? | ? | ? |
| FHSM VP-14842 | ? | 14 | ? | ? | ? | ? | ? | ? |
| FHSM VP-14843 | ? | 11 | ? | ? | ? | ? | ? | ? |
| FHSM VP-14844 | ? | 15 | ? | ? | ? | ? | ? | ? |
| *T. proriger* | | | | | | | | |
| RMM 5610 | 611** | 21** | 130** | 72* | ? | ? | ? | ? |
| CMN 51258-51263 | ? | ? | ? | 70* | ? | ? | ? | ? |
| CMN 8162 | 574 | ? | 127 | 71 | 575 | 364 | 60* | 172 |
| KUVP 5033 | 1,700* | 87* | 330* | 225 | 1,850* | 900* | 222* | 315* |
| FHSM VP-3 | 1,130 | 58 | 225 | 165 | 1,228 | 694 | 152 | 218 |
| FMNH P15144 | 1,201 | 63 | 259 | 173 | 1,343 | 761 | 84 | 239 |
| AMNH FARB 221 | 1,180* | ? | ? | 135* | 1,132* | 617* | 87 | ? |
| AMNH FARB 4909 | 610 | 42 | 143 | 78 | 695 | 416 | 71 | 138 |
| AMNH FARB 1555 | ? | ? | ? | 152 | ? | ? | ? | ? |
| USNM 6086 | 585 | ? | 142 | 79 | 650 | 373 | ? | 163 |
| USNM 8898 | 710 | 40* | 223 | ? | 935 | 565 | ? | 215 |
| YPM 1268 | ? | ? | 141 | 78 | ? | ? | ? | 130 |
| YPM 3977 | ? | 33 | ? | 82 | ? | 399 | ? | 144 |
| YPM 4002 | ? | 36 | 234 | ? | ? | ? | ? | 220 |
| YPM 3981 | ? | 57 | ? | 158 | ? | ? | ? | ? |
| KUVP 1032 | 1,212 | 57 | 268 | 170 | 1,351 | 716 | 126* | 260 |
| AMNH FARB 1585 | ? | ? | 83 | ? | ? | ? | ? | ? |
| KUVP 66129 | 506 | 19 | 129 | 63 | 553 | 345* | 47 | 120 |
| FFHM 1997-10 | 1,016 | 61 | 284 | 150 | 1,220 | 667 | ? | 251 |
| TMP 1982.050.0010 | 810 | 46 | 186 | 111 | 872 | 543 | ? | 174 |
| FMNH UR902 | ? | ? | ? | 75 | ? | ? | ? | ? |
| FMNH UR820 | ? | 54 | ? | ? | ? | ? | ? | ? |
| GSM 1 | 980 | 62 | 241 | 133 | 1,092 | 603 | ? | 223 |
| ROM 7906 | 1,005 | 53 | 256 | 144 | 1,245 | ? | ? | 235 |
| AMNH FARB 2160 | ? | 20 | ? | ? | ? | ? | ? | ? |
| AMNH FARB 1560 | ? | 41 | ? | ? | ? | ? | ? | ? |
| AMNH FARB 1592 | ? | ? | ? | 71 | ? | ? | ? | ? |
| FHSM VP-6907 | ? | 45 | ? | ? | ? | ? | ? | 165 |
| KUVP 1033 | 813 | 44 | 193 | 106 | 931 | 538 | 99 | 182 |
| KUVP 50090 | 1,300 | 49* | 272* | ? | 1,415* | 780* | 159 | 360* |
| KUVP 28705 | 615 | 31 | 138* | ? | ? | ? | ? | ? |
| KUVP 65636 | 1,180* | 56 | 149 | 150 | 1,200* | 635 | 122 | 219 |
| KUVP 1020 | ? | ? | ? | 89 | ? | ? | ? | ? |
| *T. nepaeolicus* | | | | | | | | |

(Continued)
| Table 1 (continued) | | | | | | | | |
|---|---|---|---|---|---|---|---|---|
| Specimen | A | B | C | D | E | F | G | H |
| AMNH FARB 1565 | ? | ? | ? | 78 | 660 | ? | ? | ? |
| AMNH FARB 124/134 | 717 | 19 | 176 | 92 | 828 | 444 | 85 | 180 |
| YPM 3980 | ? | ? | 181 | 110 | ? | ? | ? | ? |
| YPM 3970 | ? | ? | ? | 121 | ? | ? | ? | ? |
| YPM 3969 | ? | 25* | ? | ? | ? | ? | ? | ? |
| YPM 3974 | ? | 23 | 139 | 82* | ? | 391 | ? | 149 |
| AMNH FARB 1561 | ? | 41 | ? | ? | ? | ? | ? | ? |
| FHSM VP-7262 | ? | 44 | 175 | 106 | ? | 585* | 94 | 170 |
| FHSM VP-2209 | 851* | 44 | 201 | 133 | 1,002 | 580 | 107 | 192 |
| YPM 3979 | ? | 10 | 85 | ? | ? | 236 | ? | 83 |
| YPM 3992 | ? | ? | 99 | 46 | ? | 247 | ? | 90 |
| YPM 4000 | ? | 28 | ? | 68 | ? | 355 | ? | 135 |
| YPM 3976 | ? | 33 | ? | 109 | ? | ? | ? | ? |
| AMNH FARB 2167 | ? | ? | ? | 155* | ? | ? | ? | ? |
| *T. kansasensis* | | | | | | | | |
| FHSM VP-2295 | 650 | 27 | 154 | 82 | 724 | 404 | 72 | 130 |
| FHSM VP-78 | 378 | 14 | 75 | 43 | 440 | 251 | 41 | 81 |
| FHSM VP-2495 | ? | ? | 102 | ? | 510 | 273 | 50 | 94 |
| FHSM VP-3366 | ? | 35 | 164 | 93 | ? | 441 | ? | 164 |
| FHSM VP-9350 | ? | 11 | ? | 37 | 370 | 183 | 32 | 65 |
| FHSM VP-13742 | ? | 28* | ? | ? | 980 | 509 | 95* | 176 |
| FHSM VP-14848 | ? | ? | ? | 24 | ? | ? | ? | ? |
| FHSM VP-15631 | ? | 22 | ? | ? | 760 | ? | ? | 127* |
| FHSM VP-15632 | 360* | 16* | 82 | 45 | 414 | 240 | 39 | 71 |
| FGM V-43 | 890 | 39 | 173 | 97 | 830 | 475 | 81 | 157 |
| MCZ 1589 | ? | 20 | ? | ? | 809** | 460 | ? | ? |
| YPM 40796 | ? | ? | ? | ? | 430** | 240 | ? | ? |
| LACMNH 127815 | 650** | ? | ? | ? | 730** | 410 | ? | ? |
| TMM 40092-27 | ? | 14 | ? | ? | ? | ? | ? | ? |
| TMM 81051-64 | ? | 13 | ? | ? | ? | ? | ? | ? |
| IPB R322 | 350* | ? | 75* | 40* | 410* | 250* | ? | ? |
| FHSM VP-17206 | ? | 26 | ? | ? | ? | ? | ? | ? |
| FHSM VP-14840 | ? | 13 | ? | ? | ? | ? | ? | ? |
| FMNH PR2103 | 653 | 32 | 140* | 87 | 723 | 415 | 84 | 134 |
| FMNH UC1342 | ? | ? | ? | ? | ? | 352 | 68 | 127 |
| FHSM VP-18520 | ? | 31 | 169 | ? | ? | ? | ? | ? |

Note:
Measurements are rounded to the nearest whole millimeter. (A) Total skull length. (B) Premaxilla predental rostrum length. (C) Length between first and sixth maxillary teeth. (D) Quadrate height. (E) Lower jaw length. (F) Dentary length. (G) Dentary height. (H) Length between first and sixth dentary teeth. Measurement sources are listed in Table S1. Estimates made by the author using scale bars in the literature or due to incomplete material are indicated by a single asterisk, estimates from the literature are indicated by two asterisks, and missing measurements are indicated by question marks. TMM 1982.050.0010 is a cast of LACMNH 28964; CMN 51258 through 51263 are fragments from a single individual (*Stewart & Mallon, 2018*); AMNH FARB 124 and 134 are a skull and jaws, respectively, from a single individual (*Jiménez-Huidobro & Caldwell, 2019*); a measurement was published for CMN 8162 (B) (*Stewart & Mallon, 2018*), but it is inaccurate due to restoration of the specimen (T. Konishi, 2019, personal communication).

quadrate poorly developed or absent; (7) suprastapedial process of quadrate reaches half the length of the complete bone; (8) tympanic ala thick; (9) mandibular condyle of the quadrate mediolaterally broad; and (10) lateral crest of tympanic ala ends posteriorly near mandibular condyle (*Jiménez-Huidobro & Caldwell, 2019*).

*Tylosaurus kansasensis* is diagnosed by the following cranial characters: (1) premaxilla rostral foramina large; (2) infrastapedial process of quadrate poorly developed or absent; (3) medial ridge of quadrate diverges ventrally; (4) frontal with dorsal midline crest that is high, thin, and well-developed; (5) medial sutural flanges of frontal large, extend long distance onto parietal; (6) parietal foramen adjacent to or invading frontal-parietal suture; (7) dorsal postorbitofrontal with low rounded transverse ridge; (8) posteroventral angle of jugal is 90 degrees; (9) ectopterygoid does not contact maxilla; (10) quadrate suprastapedial process without constriction; (11) quadrate ala thick; (12) alar concavity of quadrate shallow (*Everhart, 2005*).

## Project Goals

The goals of this project were to use quantitative cladistic analysis to (1) recover growth series of *T. proriger* and *T. kansasensis/nepaeolicus*; (2) test whether total skull length (TSL) or quadrate height (QH) are appropriate proxies for relative maturity in these species; (3) test for sexual dimorphism in these species; (4) test the hypothesis that *T. kansasensis* represent juveniles of *T. nepaeolicus* (*Jiménez-Huidobro, Simões & Caldwell, 2016*); (5) test the hypothesis that two character states, the presence of a frontal midline crest and convex lateral borders of the parietal table, in *T. proriger* are paedomorphic relative to *T. nepaeolicus* (*Jiménez-Huidobro, Simões & Caldwell, 2016*); (6) test for anagenesis in these species using ontogenetic data; (7) propose revised cranial diagnoses of *T. proriger* and *T. nepaeolicus/kansasensis* within an ontogenetic context; and (8) identify conserved patterns of growth in *Tylosaurus*.

# MATERIALS AND METHODS

## Quantitative cladistic analysis

### Size-independent assessment of maturity

In fossil taxa, it is difficult to discern whether morphologically similar, but differently sized, individuals are different species or different growth stages of a single species; adults of a small species may be mistaken for juveniles of a large species, or different growth stages of a single species may be mistaken for separate species altogether (*Rozhdestvensky, 1965*; *Brinkman, 1988*; *Carr, 1999*). Furthermore, although size may help to organize individuals into general categories (e.g., "juveniles," "subadults," "adults"), it is not possible to precisely determine the maturity of the individuals within each of these categories using size alone (i.e., the biggest individual is not necessarily the most mature) (*Brinkman, 1988*; *Carr, 2020*).

To solve this issue, *Brinkman (1988)* suggested the identification of size-independent ontogenetically variable characters (i.e., morphological features such as bone shape and texture, suture shape and closure, degree of ossification, etc.). This does not mean that size is completely uninformative, but that it is simply that more information is needed to

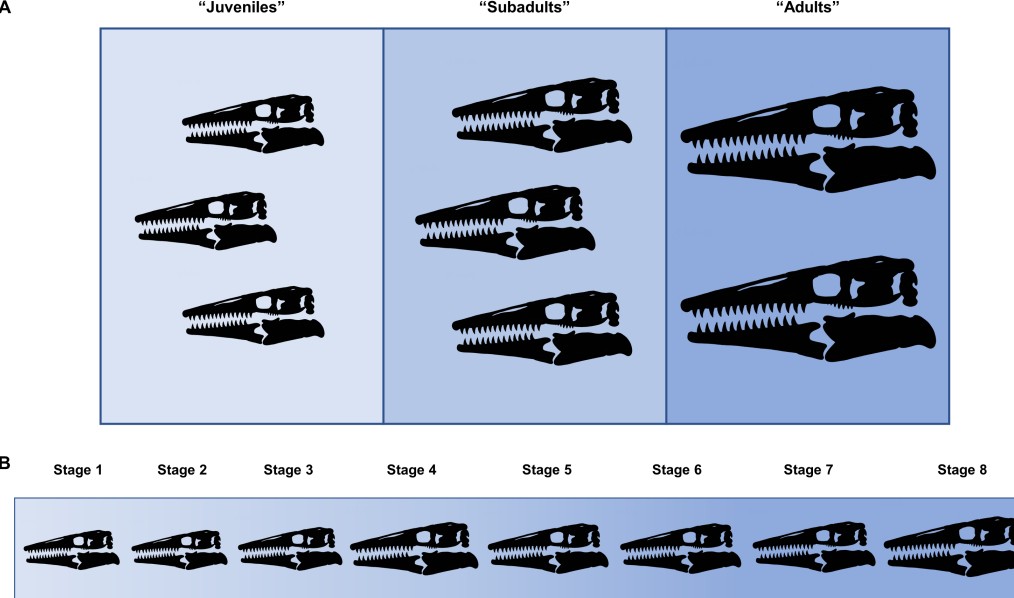

**Figure 1 Comparison between hypothetical low-resolution and high-resolution growth series.** (A) In a low-resolution growth series, multiple individuals are grouped into vague sets. (B) In a high-resolution growth series, each growth stage only has a single individual.

accurately assess the relative maturities of individuals through character congruence (i.e., multiple lines of evidence) instead of size alone, which is variable (*Brinkman, 1988*; *Carr, 2020*). Therefore, both size-dependent and size-independent characters must be considered when proposing hypotheses of growth.

### Cladistic analysis of growth

Ontogeny, like evolution, consists of a hierarchical accumulation of changes over time (*Brochu, 1996*). Thus, in the same way that the evolutionary relationships between taxa are recovered, cladistic analysis can be used to identify the relative maturity of specimens drawn from a sample of a single species. This method allows size-dependent and size-independent data to be combined to recover a high-resolution growth series that is more precise than simply grouping multiple individuals into imprecise sets such as "juveniles," "subadults," and "adults" (Fig. 1).

Separate character matrices were compiled for *T. proriger* and *T. kansasensis/ nepaeolicus* (Data S1 and S2). FHSM VP-14845, which is only identifiable to *Tylosaurus* sp., was included in both datasets given that it was found between the two species stratigraphically and could be referable to either taxon (*Konishi, Jiménez-Huidobro & Caldwell, 2018*). Character states with the immature condition were coded with zeroes and increasingly mature states were coded with progressively higher numbers. Multistate characters were coded for characters that are not binary (three or more states), and all characters were run unordered and equally weighted. A hypothetical embryo, scored with all zeroes, was added as the analog of the outgroup in each dataset to polarize the characters, since an embryo expresses the least mature condition of all character states and

because no single juvenile specimen is coded with all zeroes (*Brochu, 1996*; *Carr & Williamson, 2004*; *Frederickson & Tumarkin-Deratzian, 2014*; *Carr, 2020*).

Growth stages were defined corresponding to the nodes on the ontogram, and the growth characters that diagnose each stage were the unambiguously optimized synontomorphies (shared growth characters; *Frederickson & Tumarkin-Deratzian, 2014*; *Carr, 2020*). Growth characters that were unambiguously optimized on the branches to individual specimens were considered individual variation. Following the method of *Carr & Williamson (2004)*, *Frederickson & Tumarkin-Deratzian (2014)*, and *Carr (2020)*, an artificial adult was added a posteriori to identify the most mature specimen of each taxon. The artificial adult was scored with the character states optimized at the most mature node (i.e., the node supported by the most synontomorphies). Should the analysis with the artificial adult fail to recover a single most mature specimen, the individual specimen with the most growth changes—that is, with the greatest number of synontomorphies—was considered the most mature (Fig. 2).

## Compilation and analysis of the data matrices

This project makes use of data drawn from 79 specimens housed in several North American institutions, as well as one in Japan and one in Germany (Table S1); of those, 50 were studied first-hand at the Field Museum of Natural History (Chicago, IL), Fryxell Geology Museum (Rock Island, IL), American Museum of Natural History (New York, NY), Sternberg Museum of Natural History (Hays, KS), and University of Kansas Museum of Natural History (Lawrence, KS). All others were scored from descriptions and measurements in the literature, and photographs online or in the primary literature (exact sources for coding each specimen are listed in Table S1; analyses run including only specimens studied first-hand are shown in Figs. S1A–S1C). The total numbers of specimens scored for each taxon are as follows: five *Tylosaurus* sp.; 39 *T. proriger*; 21 *T. kansasensis*; and 14 *T. nepaeolicus*. Several specimens of each taxon (i.e., "wildcard" specimens which resulted in multiple equally parsimonious ontograms) were removed from the final analyses due to incomplete or redundant coding following the method of *Carr (2020)* (Table S2), and any characters that were not scored for more than a single specimen were excluded from the analyses.

Hypothetical growth characters were identified by the author and in the literature, and include both diagnostic characters (*Bell, 1997*; *Jiménez-Huidobro & Caldwell, 2019*) and characters explicitly proposed to be ontogenetically variable (*Harrell & Martin, 2015*; *Jiménez-Huidobro, Simões & Caldwell, 2016*; *Stewart & Mallon, 2018*). Characters are described in detail in Data S3, and measurements and tooth counts are listed in Tables 1 and 2, respectively. A total of 59 characters were identified, which includes two measures of size (TSL and QH), seven proportional characters, 19 size-independent characters, and 30 phylogenetic characters (e.g., characters that are either diagnostic for one of the species or that are purportedly ontogenetically variable and are also phylogenetic characters of *Bell (1997)*) (Data S3; see Fig. S2 for exemplars of select morphological characters and their states). Of the phylogenetic characters, 11 were not figured in the sources that identify them, and so they could not be identified with certainty nor scored

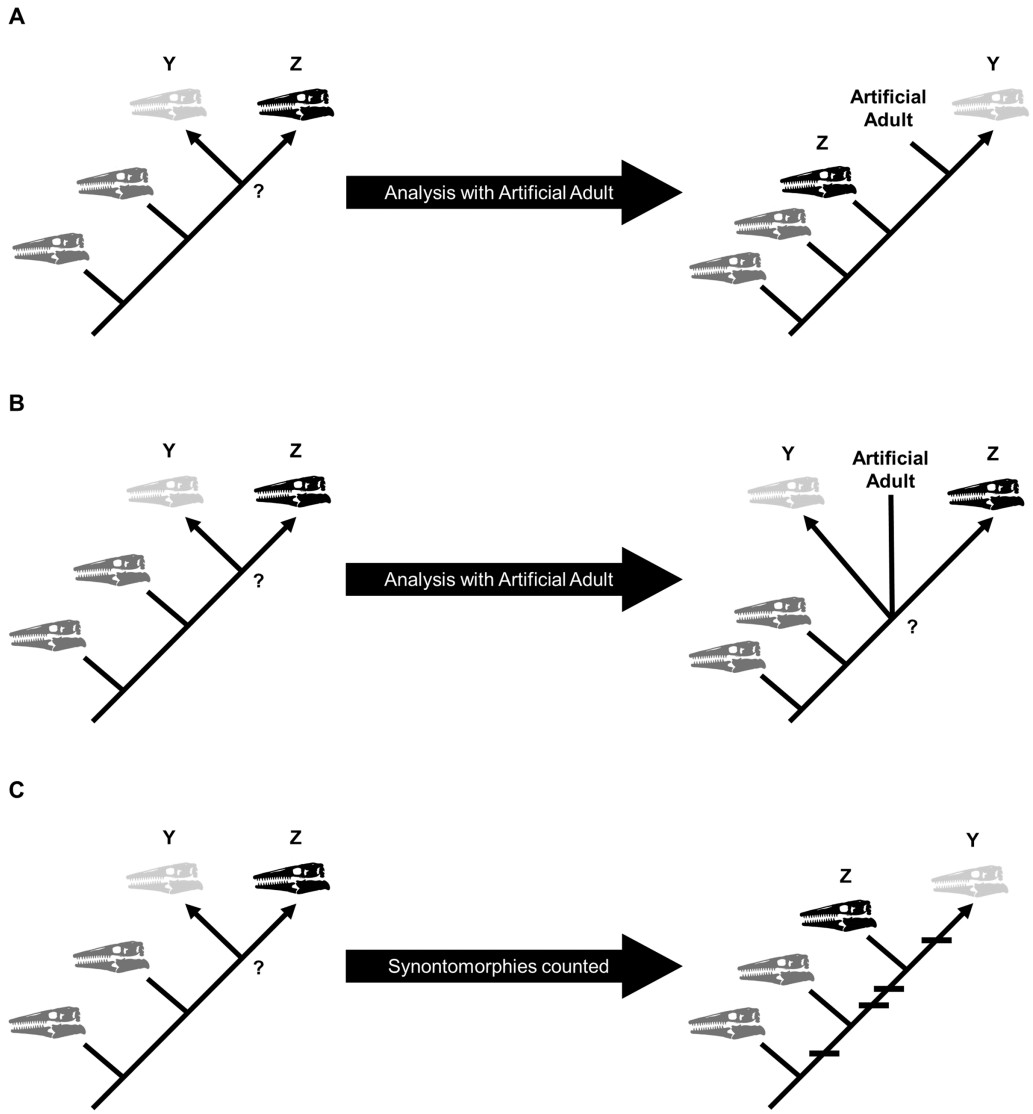

**Figure 2 Possible scenarios when determining the most mature individual.** In each scenario, a cladistic analysis has recovered hypothetical specimens "Y" (light gray) and "Z" (black) at the terminus of the ontogram. The most mature individual(s) is indicated by an arrowhead. (A) The analysis with an artificial adult is successful; the artificial adult is recovered closest to specimen "Y," indicating that it is the most mature. (B) The analysis with the artificial adult fails to recover a single most mature specimen; the artificial adult is not closer to specimen "Y" or specimen "Z." (C) Should the analysis with the artificial adult fail, the specimen with the most accumulated growth changes (synontomorphies) is considered the most mature; in this scenario, the most mature individual is specimen "Y," with a total of four synontomorphies.

consistently by the author (noted in Data S3); therefore, while they are included in the data matrices and the character list, they were excluded from all of the analyses, and any codes for those characters are from the literature.

Proportions were calculated and rounded to the nearest whole percent, and those that seemed to show variation due to growth (e.g., a difference of 3% or more between specimens of purportedly different maturities) were coded using specimens referred to by
**Table 2 Known tooth counts of specimens included in this project.**

| Specimen | Maxillary teeth | Dentary teeth | Pterygoid teeth |
|---|---|---|---|
| *T. proriger* | | | |
| CMN 8162 | 13 | 13 | ? |
| FHSM VP-3 | 13 | 13 | ? |
| FMNH P15144 | 13 | 14 | 10 |
| AMNH FARB 4909 | ? | 13 | 10 |
| KUVP 1032 | 13 | 13 | 10 |
| KUVP 66129 | ? | 12 | ? |
| FFHM 1997-10 | 13 | 13 | ? |
| KUVP 1033 | 13 | 13 | ? |
| KUVP 28705 | 13 | ? | 10 |
| KUVP 65636 | 12 | 13 | ? |
| *T. nepaeolicus* | | | |
| AMNH FARB 124/134 | 13 | 14 | 8, 9 |
| FHSM VP-7262 | 12 | 12 | 10, 9 |
| FHSM VP-2209 | 13 | 14 | ? |
| *T. kansasensis* | | | |
| FHSM VP-2295 | 13 | 13 | ? |
| FHSM VP-78 | ? | 12 | ? |
| FHSM VP-2495 | ? | 13 | ? |
| FHSM VP-3366 | ? | 11–12 | ? |
| FHSM VP-9350 | ? | 13 | ? |
| FHSM VP-13742 | ? | 13 | ? |
| FHSM VP-15632 | 12 | 15, 13 | ≥11 |
| FGM V-43 | 13 | 13, 12 | 8 |
| IPB R322 | 12 | ? | ? |
| FMNH PR2103 | 13 | 10, 12 | 13, 11 |
| FMNH UC1342 | ? | 13 | ? |

Note:
Missing counts are indicated by question marks. If tooth counts were available for both left and right bones, the number of teeth on the left bone is listed first.

the literature as "juveniles" (e.g., CMN 8162, RMM 5610) and "adults" (e.g., AMNH FARB 4909, FHSM VP-3). Size characters (TSL and QH) were rounded to the nearest whole millimeter and states were coded as roughly equal bins spanning the known range of sizes of both taxa (Table 1; Data S3).

Continuous variables, such as size, are potentially problematic in phylogenetic analyses for several reasons, namely, that variations due to ontogeny or sexual dimorphism may obscure evolutionary relationships, and it is difficult to determine the ancestral state of size characters, or partition continuous variables in general, without introducing personal biases (*Rae, 1998*; *Simões et al., 2016*). However, in this work (a specimen-by-specimen analysis of ontogeny), these concerns are irrelevant; variation in topology due to ontogeny and sexual dimorphism is exactly what is sought by this type of analysis,

and unlike in phylogenetic studies, the ancestral states of size characters are not ambiguous or arbitrary, given that it is not unreasonable to assume that animals will generally get larger as they mature, and the analysis itself tests if that hypothesis is defensible or not through character congruence. To test the effect of the size characters on ontogram topology, the analyses were also run excluding them (Figs. S1D–S1I).

Most phylogenetic character states were coded as they are in *Bell (1997)*, and ontogenetic characters were coded according to literature descriptions or naïvely according to patterns uncovered in this project (i.e., the state seen in individuals proposed to be immature by other work (e.g., FHSM VP-14845, RMM 5610, CMN 8162, FHSM VP-15632) was coded as the less developed state, and the state seen in individuals proposed to be more mature by other work (e.g., AMNH FARB 1555, FHSM VP-3, AMNH FARB 124/134) was coded as the more developed state). Data matrices were compiled in Mesquite (*Maddison & Maddison, 2018*) and analyzed in TNT (*Goloboff & Catalano, 2016*) and PAUP (*Swofford, 2003*). TNT was used to recover the ontogram topology and number of most parsimonious trees using a new technology search followed by a traditional search; the topology was then loaded as a constraint into PAUP, which recovered the synontomorphies using branch-and-bound searches.

## Testing congruence between size and maturity

Size alone is often not a reliable indicator of relative maturity (*Rozhdestvensky, 1965*; *Brinkman, 1988*; *Brochu, 1996*; *Carr, 2020*). To test this hypothesis in mosasaurs, once the growth series were recovered, the congruence between size (TSL) and maturity in each taxon was tested using the method of *Frederickson & Tumarkin-Deratzian (2014)* and *Carr (2020)*, where the growth stages and TSL measurements for each specimen were converted into ranks and then analyzed in SPSS (*IBM Corp., 2019*) using a Spearman rank-order correlation test. If size and maturity are congruent (i.e., larger individuals tend to be more mature), the correlation will be positive and statistically significant ($p < 0.05$). Because mosasaur skulls are not always complete enough for an accurate measurement or estimate of TSL, the same method was used to test the congruence between QH and maturity. The normality of the growth ranks, size ranks, and measurement data were tested using a Shapiro–Wilk test.

## Testing sexual dimorphism and taxon validity

The ontogram recovered by a cladistic analysis can be used to test for the presence of sexual dimorphism (*Frederickson & Tumarkin-Deratzian, 2014*). If no evidence for sexual dimorphism is recovered, the ontogram will be linear (i.e., one specimen per node; Fig. 3A). If, however, sexual dimorphism is present, the ontogram hypothetically will bifurcate (i.e., a single node will have two groups of multiple specimens) into two groups of specimens, corresponding to each sex, after one or more juvenile stages (Figs. 3B and 3C). It is also possible that the ontogram is linear and sexual dimorphism is instead recovered as two homologous sets of individual variation (Fig. 3D). Following the reasoning of *Frederickson & Tumarkin-Deratzian (2014)*, if a bifurcation (or set of

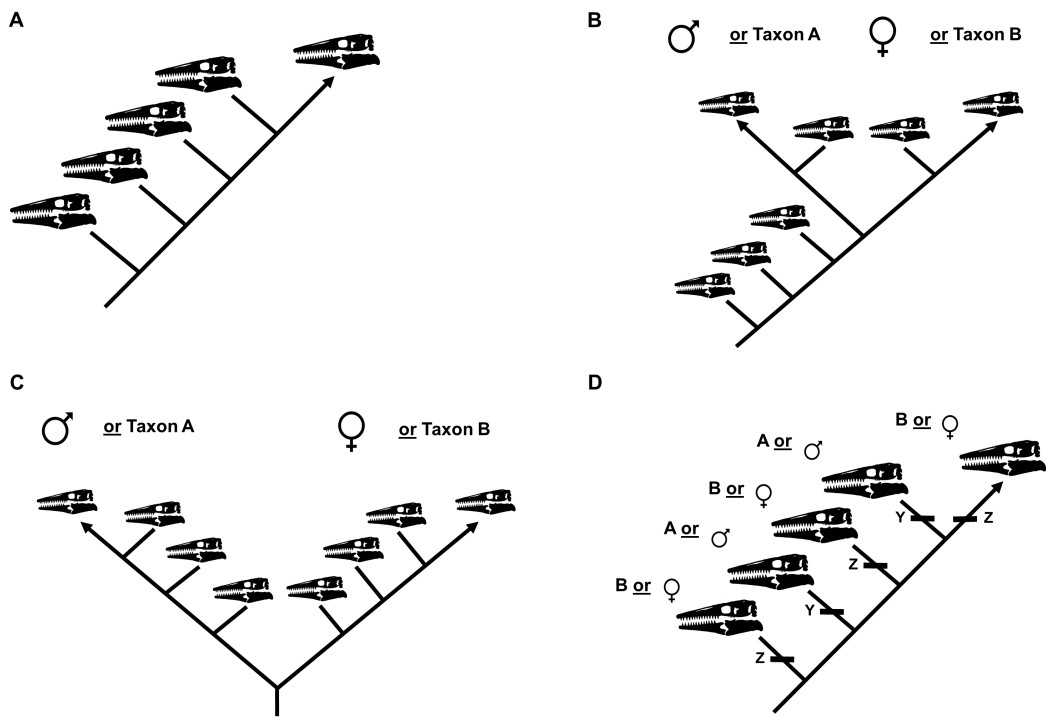

**Figure 3 Summary of potential outcomes for the growth series recovered by the cladistic analysis.**
(A) The specimens included in the analysis represent a single taxon without sexual dimorphism.
(B) The specimens in the analysis represent either a single taxon that is sexually dimorphic or two separate taxa with morphologically similar juveniles. (C) The specimens in the analysis represent either a single taxon that is sexually dimorphic with an oversampling of adults or two separate taxa. (D) The analysis recovers two or more groups of specimens defined by shared instances of individual variation; these groups could represent different taxa or sexual dimorphism.

individual variation) represents sexual dimorphism, each sex should (1) independently develop a shared sequence of growth changes, since they are the same taxon, in addition to (2) developing unique morphological features that are hypothetically used for sexual display.

The growth series will also be used to test the validity of specimens assigned to each taxon. If specimens assigned to the taxon actually represent two or more different species, the ontogram hypothetically will bifurcate into two or more groups (Figs. 3B and 3C) or it will be linear and recover two or more groups defined by shared sets of individual variation (Fig. 3D).

## Test of synonymy between *T. kansasensis* and *T. nepaeolicus*

To test the hypothesis that *T. kansasensis* are juveniles of *T. nepaeolicus*, a single matrix including specimens of both taxa was constructed. Summaries of potential results are shown in Fig. 4. This is not the first study that has used a cladistic analysis of ontogeny to test a hypothesis regarding synonymy; *Longrich & Field (2012)* used the same approach to test, and reject, the hypothesis that specimens of the genus *Torosaurus* represent adults of another genus of North American horned dinosaur, *Triceratops*.

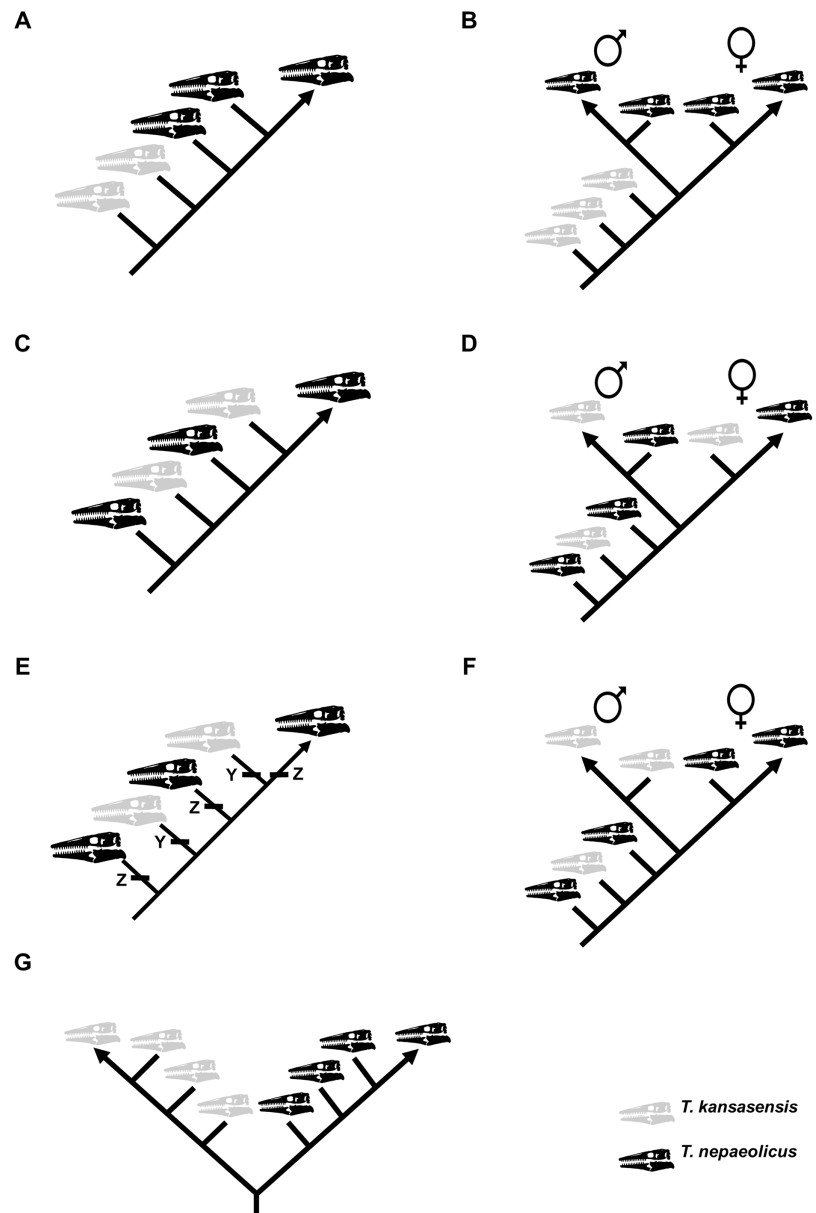

**Figure 4 Summary of potential outcomes for the analysis of the data matrix including *Tylosaurus kansasensis* and *Tylosaurus nepaeolicus*.** Hypothetical *T. nepaeolicus* specimens are represented by black skulls and hypothetical *T. kansasensis* specimens are represented by gray skulls. (A) If the current hypothesis is supported and *T. kansasensis* are juveniles of *T. nepaeolicus* (*Jiménez-Huidobro, Simões & Caldwell, 2016*), most or all *T. kansasensis* specimens will be recovered as less mature than most or all *T. nepaeolicus* specimens. (B) If *T. kansasensis* are juveniles of *T. nepaeolicus*, and the taxon is sexually dimorphic, most or all *T. kansasensis* specimens will be recovered as less mature than most or all *T. nepaeolicus* specimens and before the onset of sexual maturity (represented by a bifurcation in the ontogram). (C) If *T. kansasensis* and *T. nepaeolicus* are the same taxon but neither is necessarily all adults nor all juveniles, and sexual dimorphism is absent, the specimens will be interspersed with each other on the ontogram. (D) If *T. kansasensis* and *T. nepaeolicus* are the same taxon but neither is necessarily all adults nor all juveniles, and sexual dimorphism is present, the specimens will be interspersed with each other on the ontogram and on both branches after the onset of sexual maturity. (E) The ontogram is linear with specimens of both taxa interspersed with each other, but identical individual variations are unambiguously optimized in several specimens of one taxon and not along the main axis or in specimens of the other taxon; in this case, two groups are recovered and they may represent two taxa or sexual

**Figure 4** (continued)
dimorphism. (F) If *T. kansasensis* and *T. nepaeolicus* are opposite sexes of the same taxon, the ontogram will bifurcate with specimens of *T. kansasensis* on one branch, *T. nepaeolicus* on the other branch, and a mix of specimens near the root. (G) If *T. kansasensis* and *T. nepaeolicus* are two different taxa, the ontogram will bifurcate at or near the root with all the *T. kansasensis* specimens on one branch and all the *T. nepaeolicus* specimens on the other; this could also represent sexual dimorphism with an oversampling of adults in which specimens of *T. kansasensis* represent one sex and specimens of *T. nepaeolicus* represent the other.                               

## Test of anagenesis and heterochrony in *Tylosaurus*

Anagenesis–evolution within a single lineage (i.e., without branching into multiple new clades) over time–has been studied in several nonavian dinosaur taxa as a mechanism for producing species diversity, particularly in ceratopsians and tyrannosaurs (*Horner, Varricchio & Goodwin, 1992*; *Scannella et al., 2014*; *Carr et al., 2017*; *Wilson, Ryan & Evans, 2020*). In order for anagenesis to be defensible, the taxa in question must meet the following criteria: (1) they do not overlap stratigraphically; (2) they have a close phylogenetic relationship; (3) some specimens have intermediate morphology; and (4) they inhabited the same location (*Carr et al., 2017*; *Wilson, Ryan & Evans, 2020*).

No previous study has proposed anagenesis as a mechanism of speciation in mosasaurs. Because of the large sample size and potential for high-resolution growth series, they are an ideal taxon for testing hypotheses of evolutionary processes, particularly anagenesis (*Carr et al., 2017*). In this project, the novel hypothesis that the *Tylosaurus* of the WIS (*T. proriger* and *T. kansasensis/nepaeolicus*) are a single, anagenetic lineage will be tested. The two *Tylosaurus* taxa meet each criterion for anagenesis outlined above: (1) *T. kansasensis/nepaeolicus* and *T. proriger* do not overlap stratigraphically; (2) they are sister taxa (*Jiménez-Huidobro & Caldwell, 2019*); (3) some specimens have intermediate morphology (e.g., the quadrate infrastapedial process is absent or weak in *T. kansasensis* and *T. nepaeolicus*, and it is always present and well-developed in *T. proriger*); and (4) they both lived in the WIS (it is important to note, however, that the WIS was connected to the oceans; although fossils of these species have thus far never been found outside the WIS, the possibility of them occasionally leaving the WIS to recolonize elsewhere cannot be ruled out).

If the cladistic analysis of growth based on the dataset including specimens of *T. kansasensis* and *T. nepaeolicus* supports their synonymy, then a single data matrix including specimens of all three taxa (i.e., *T. kansasensis*, *T. nepaeolicus* and *T. proriger*) will be compiled and analyzed. If the hypothesis of anagenesis is supported, and speciation in WIS *Tylosaurus* was driven by peramorphy (extension or acceleration of growth; *Reilly, Wiley & Meinhardt, 1997*), then the ontogram will show a progression from *T. kansasensis/nepaeolicus* to *T. proriger*, and if speciation was driven by paedomorphy (truncation or deceleration of growth), the ontogram will either show a progression from *T. proriger* to *T. kansasensis/nepaeolicus* or a progression from *T. kansasensis/nepaeolicus* to *T. proriger* that includes many character reversals. If anagenesis is not supported,

specimens of both taxa will be interspersed with one another on the ontogram or the ontogram will bifurcate basally (Fig. 3C).

Furthermore, testing for anagenesis using ontogenetic data allows for another hypothesis to be tested: heterochrony as a driver of evolution in mosasaurs. Heterochrony is differences in the timing of developmental events (i.e., the developmental consequences of a truncation, extension, acceleration, or deceleration of growth in one taxon relative to another) that produce the morphological differences between a descendent taxon from its ancestor (*Reilly, Wiley & Meinhardt, 1997*). If heterochrony is an evolutionary mechanism in *Tylosaurus*, and the *Tylosaurus* species of the WIS are a single anagenetic lineage, then a cladistic analysis of growth will recover the specific developmental changes that produced *T. proriger*—the descendent—from *T. kansasensis/nepaeolicus*—the ancestor.

Finally, *Jiménez-Huidobro, Simões & Caldwell (2016)* also proposed that the presence of a frontal dorsal midline crest and convex lateral borders of the parietal table are paedomorphic in *T. proriger* relative to *T. nepaeolicus* because of the purported absence of the frontal crest and straight borders of the parietal table in mature *T. nepaeolicus*. These hypotheses were tested here by comparing the growth patterns for that trait between both species; if these characters in *T. proriger* are paedomorphic relative to *T. nepaeolicus*, then they will be the same (i.e., crest present and borders convex) in all *T. proriger* specimens and in immature *T. nepaeolicus*, and different (i.e., crest absent and borders straight) in mature *T. nepaeolicus*.

## RESULTS

### Growth series of *T. proriger*

One ontogram was recovered with a length of 82 steps, consistency index (CI) of 0.65, homoplasy index (HI) of 0.35, retention index (RI) of 0.76, and rescaled consistency index (RC) of 0.49 (Fig. 5). The topology was assessed using a Bremer decay index approach, and resolution was lost after the addition of one step. A total of 17 growth stages were identified; the analysis with the artificial adult and all 23 specimens did not recover a single most mature specimen, but a second analysis with the artificial adult which only included the three most mature specimens (i.e., those that were sister to the artificial adult in the analysis with all 23 specimens: FMNH P15144, FHSM VP-3, and AMNH FARB 1555) identified FHSM VP-3 as the most mature individual. Optimized synontomorphies that support each node (i.e., growth stage) are listed in Table 3, and character states that were unambiguously optimized as individual variation are listed in Table S3. The following growth stages are recovered based on the unambiguously optimized synontomorphies that support each node on the ontogram:

**Stage 1.** This stage is recovered as sister to the other specimens (exemplar: *Tylosaurus* sp. neonate FHSM VP-14845).

**Stage 2.** The QH is between 50 and 99 mm and the mandibular condyle of the quadrate is completely ossified (exemplar: CMN 51258-51263).

**Stage 3.** The quadrate tympanic ala is thick (Fig. 6) (exemplar: AMNH FARB 1592).

**Stage 4.** The quadrate alar concavity is shallow (Fig. 6) (exemplar: FMNH UR902).

**Stage 5.** The occipital condyle is completely ossified (exemplar: AMNH FARB 2160).

**Stage 6.** The foramina on the premaxillary rostrum are small (Fig. 7) (exemplar: RMM 5610).

**Stage 7.** The premaxilla-maxilla suture is m-shaped (Fig. 8) and the mandibular condyle of the quadrate is rounded (Fig. 6) (exemplar: KUVP 66129).

**Stage 8.** The infrastapedial process of the quadrate is rounded (Fig. 6) (exemplar: CMN 8162).

**Stage 9.** The QH is ≥13% TSL and the dentary is deep (i.e., ≤6 times longer than tall) (exemplars: AMNH FARB 4909, KUVP 28705, KUVP 1033, and TMP 1982.050.0010). At this stage, the exemplar specimens share a distance between the first and sixth dentary teeth that is ≤23% TSL and ≤35% dentary length; KUVP 28705, KUVP 1033, and TMP 1982.050.0010 share a reversal to foramina on the premaxillary rostrum that are large and frontal medial suture flanges that are large (Fig. 9); and KUVP 1033 and TMP 1982.050.0010 share a TSL that is between 800 and 999 mm.

**Stage 10.** The frontal posterolateral processes are thick (Fig. 10) and the dorsal ridge on the predental process of the dentary is present (Fig. 11) (exemplars: USNM 6086 and USNM 8898).

**Stage 11.** This stage is diagnosed by a TSL that is between 1000 and 1499 mm, a QH that is between 150 and 199 mm, and a dentary length that is ≤55% lower jaw length (exemplar: FFHM 1997-10).

**Stage 12.** The premaxillary rostrum is distinctly knobbed (Fig. 7) (exemplar: KUVP 50090).

**Stage 13.** The distance between the first and sixth dentary teeth is ≤23% TSL (exemplar: KUVP 1032).

**Stage 14.** The quadrate suprastapedial process is thick (Fig. 6) (exemplars: ROM 7906, GSM 1, and AMNH FARB 221). At this stage, the exemplar specimens share a distance between the first and sixth maxillary teeth that is ≥25% TSL and a reversal to a QH between 100 and 149 mm.

**Stage 15.** The distance between the first and sixth dentary teeth is ≤35% dentary length, there is a reversal to a dentary length that is between 60% and 56% lower jaw length, and the coronoid posteroventral process is present and fan-like (Fig. 12) (exemplar: FMNH P15144).

**Stage 16.** This stage is not unambiguously diagnosed by any character states, but the node is ambiguously supported by a reversal to a premaxilla-maxilla suture that is u-shaped and a deep dentary (exemplar: AMNH FARB 1555).

**Stage 17.** This stage is diagnosed by a reversal to a quadrate alar concavity that is deep (Fig. 6) (exemplar: FHSM VP-3).

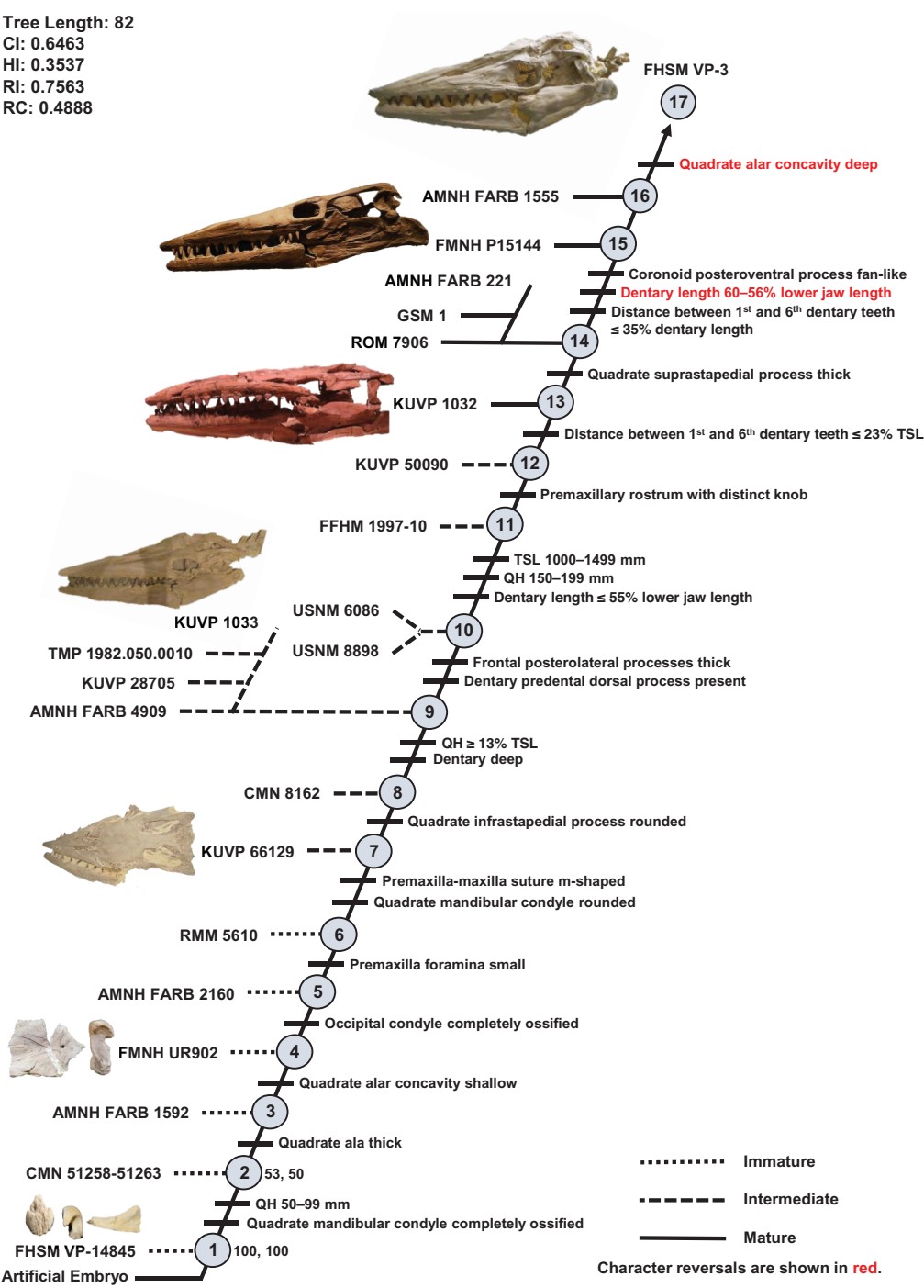

Tree Length: 82
CI: 0.6463
HI: 0.3537
RI: 0.7563
RC: 0.4888

FHSM VP-3
17
Quadrate alar concavity deep

AMNH FARB 1555

FMNH P15144
15
Coronoid posteroventral process fan-like
Dentary length 60–56% lower jaw length

AMNH FARB 221
GSM 1
Distance between 1st and 6th dentary teeth
≤ 35% dentary length
ROM 7906

Quadrate suprastapedial process thick
KUVP 1032

Distance between 1st and 6th dentary teeth ≤ 23% TSL
KUVP 50090

Premaxillary rostrum with distinct knob
FFHM 1997-10

TSL 1000–1499 mm
QH 150–199 mm
Dentary length ≤ 55% lower jaw length

KUVP 1033
USNM 6086
TMP 1982.050.0010
USNM 8898
KUVP 28705
AMNH FARB 4909

Frontal posterolateral processes thick
Dentary predental dorsal process present

QH ≥ 13% TSL
Dentary deep
CMN 8162

Quadrate infrastapedial process rounded
KUVP 66129

Premaxilla-maxilla suture m-shaped
Quadrate mandibular condyle rounded
RMM 5610

Premaxilla foramina small
AMNH FARB 2160

Occipital condyle completely ossified
FMNH UR902

Quadrate alar concavity shallow
AMNH FARB 1592

Quadrate ala thick
CMN 51258-51263
2
53, 50

QH 50–99 mm
Quadrate mandibular condyle completely ossified
FHSM VP-14845
1
100, 100
Artificial Embryo

············· Immature
----- Intermediate
——— Mature

Character reversals are shown in red.

**Figure 5 Ontogram of one *Tylosaurus* sp. specimen and 22 *Tylosaurus proriger* specimens based on a quantitative cladistic analysis.** The ontogram is a single tree and tree statistics are summarized in the upper left. Character states that define each growth stage are shown along the main branch, and the exemplar specimens are to the left of the main branch; the most mature individual, identified by the analysis with an artificial adult, is indicated by an arrow. The encircled numbers on the nodes are the growth stages, and the numbers to the right of them are the bootstrap and jackknife values, respectively (1,000 replicates, <50% not shown). Unambiguous character reversals are shown in red. "Immature" specimens were recovered in the lower third of the tree, "intermediate" specimens were recovered in the middle third of the tree, and "mature" specimens were recovered in the upper third of
**Table 3** Optimized synontomorphies supporting the growth stages of *Tylosaurus proriger*.

| Growth stage | Unambiguous | Ambiguous |
|---|---|---|
| 1 | n/a | n/a |
| 2 | QH between 50 and 99 mm, quadrate mandibular condyle ossified | TSL between 400 and 800 mm, quadrate ala rim distinct, coronoid posteroventral process present as bump |
| 3 | Quadrate tympanic ala thick** | None |
| 4 | Quadrate alar concavity shallow | None |
| 5 | Occipital condyle ossified | Quadrate suprastapedial process intermediate length**, quadrate suprastapedial process not curved medially* |
| 6 | Premaxillary rostrum foramina small | None |
| 7 | Premaxilla-maxilla suture m-shaped, quadrate mandibular condyle rounded | None |
| 8 | Quadrate infrastapedial process rounded | Premaxillary rostrum ≥5% TSL, parietal nuchal fossa present |
| 9 | QH ≥13% TSL, dentary deep | Jugal posteroventral process present* |
| 10 | Frontal posterolateral processes thick*, dorsal ridge of dentary predental process present | Pterygoid ectopterygoid process thick, coronoid anterolateral notch present and shallow |
| 11 | TSL between 1,000 and 1,499 mm, QH between 150 and 199 mm, dentary length ≤55% lower jaw length | None |
| 12 | Premaxillary rostrum distinctly knobbed | None |
| 13 | Distance between 1st and 6th dentary teeth ≤23% TSL | None |
| 14 | Quadrate suprastapedial process thick | **Dentary slender** |
| 15 | Distance between 1st and 6th dentary teeth ≤35% dentary length, **dentary length between 60% and 56% lower jaw length**, coronoid posteroventral process present and fan-like | None |
| 16 | None | **Premaxilla-maxilla suture u-shaped**, dentary deep |
| 17 | **Quadrate alar concavity deep** | None |

Note:
    Reversals are bold, phylogenetic characters are indicated by an asterisk, and characters that are purportedly diagnostic of *T. proriger* are indicated by two asterisks.

### Growth Series of *T. kansasensis* and *T. nepaeolicus*

One ontogram was recovered with a length of 87 steps, a CI of 0.59, an HI of 0.41, an RI of 0.62, and an RC of 0.36 (Fig. 13). The tree topology was assessed using a Bremer decay index approach, and resolution was lost after the addition of one step. A total of 12 growth stages were identified; the analysis with the artificial adult and all 19 specimens recovered YPM 3970 and FHSM VP-2209 as the most mature individuals (a comparison of the number of growth changes could not determine which of these two specimens is the most mature). Notably, although the holotype of *T. nepaeolicus* is recovered as more mature (stage 9) than the holotype of *T. kansasensis* (stage 8), there are no unambiguously optimized synontomorphies that distinguish them (Fig. 13; Table 4).

Optimized synontomorphies that support each growth stage are listed in Table 4, and character states that were unambiguously optimized as individual variation are listed in Table S3. The following growth stages are recovered based on the unambiguously optimized synontomorphies that support each node on the ontogram:

**Stage 1.** This stage is recovered as sister to the other specimens (exemplar: *Tylosaurus* sp. neonate FHSM VP-14845).

**Stage 2.** This stage is not unambiguously diagnosed by any character states, but the node is ambiguously supported by a quadrate tympanic ala that is thick and shallow (exemplar: *T. kansasensis* FHSM VP-17206).

**Stage 3.** The premaxilla-maxilla suture is u-shaped (Fig. 8) (exemplars: *T. kansasensis* FHSM VP-9350 and *T. kansasensis* FHSM VP-2495). At this stage, the exemplar specimens share a deep dentary.

**Stage 4.** The foramina on the premaxillary rostrum are small (Fig. 7) and the quadrate mandibular condyle is rounded (exemplar: *T. kansasensis* FHSM VP-78).

**Stage 5.** The QH is ≥13% TSL, the quadrate ala rim is defined, and the dorsal ridge on the predental process of the dentary is present (Fig. 11) (exemplar: *T. kansasensis* FHSM VP-15632).

**Stage 6.** The QH is between 50 and 99 mm and the quadrate mandibular condyle is completely ossified (exemplar: *T. kansasensis* FHSM VP-3366, *T. kansasensis* FHSM VP-18520, and *T. nepaeolicus* FHSM VP-7262). At this stage, the exemplar specimens share a decrease in dentary teeth (from 13 to 12).

**Stage 7.** The foramina on the premaxillary rostrum reverse from small to large (exemplar: *T. kansasensis* FHSM VP-15631).

**Stage 8.** The posteroventral angle of the jugal is obtuse and the coronoid posteroventral process is present as a bump (Fig. 12) (exemplar: *T. kansasensis* holotype FHSM VP-2295).

**Stage 9.** This stage is not unambiguously diagnosed by any character states, but the node is ambiguously supported by parietal lateral borders that are straight (exemplar: *T. nepaeolicus* holotype AMNH FARB 1565).

**Stage 10.** The quadrate suprastapedial process is thick (Fig. 6) and the coronoid anterolateral notch is present and shallow (exemplars: *T. kansasensis* FMNH PR2103, *T. kansasensis* FGM V-43, and *T. nepaeolicus* AMNH FARB 2167). At this stage, the exemplar specimens share a quadrate suprastapedial process that is not curved medially, and FGM V-43 and AMNH FARB 2167 share a quadrate suprastapedial process that is long.

**Stage 11.** The premaxillary rostrum is distinctly knobbed (Fig. 7), the frontal posterolateral processes are thick (Fig. 9), and there is an increase in dentary teeth (from 13 to 14) (exemplars: *T. nepaeolicus* YPM 3974 and *T. nepaeolicus* AMNH FARB 124/134). At this stage, the exemplar specimens share an absence of the parietal nuchal fossa and a distance between the first and sixth dentary teeth that is greater than 35% dentary length.

**Stage 12.** This stage is diagnosed by a QH that is between 100 and 149 mm (exemplars: *T. nepaeolicus* YPM 3970 and *T. nepaeolicus* FHSM VP-2209).

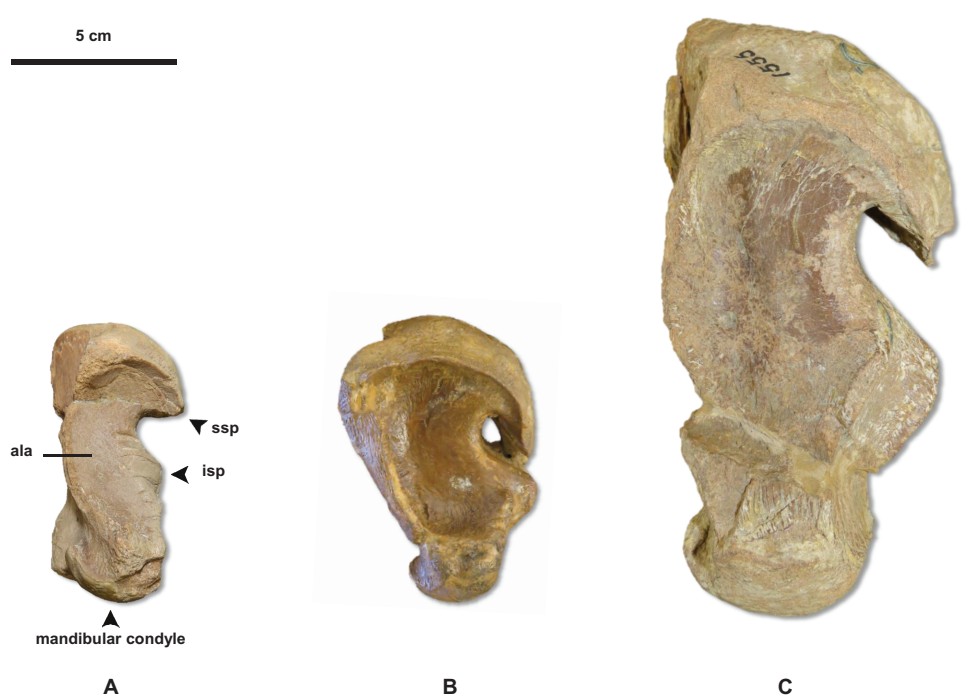

5 cm

**Figure 6 Variation in *Tylosaurus* quadrate shape.** (A) *T. proriger* FMNH UR902. (B) *T. proriger* AMNH FARB 4909. (C) *T. proriger* AMNH FARB 1555. The infrastapedial process is either broadly pointed (A) or expanded, rounded, and semicircular (B and C). The suprastapedial process is either slender (B) or robust (A and C). The tympanic ala is either thick (A) or thin (B and C) and the alar concavity is either deep (B) or shallow (A and C). Distinct deflection of the mandibular condyle is either present (A) or absent (B and C). Abbreviations: **isp**, infrastapedial process of the quadrate; **ssp**, suprastapedial process of the quadrate. The photograph of FMNH UR902 has been inverted to face left.

## Analysis Including *T. kansasensis*, *T. nepaeolicus*, and *T. proriger*

Because the synonymy of *T. kansasensis* and *T. nepaeolicus* is supported, a data matrix including all three *Tylosaurus* taxa was analyzed (Data S4). Two most parsimonious trees were recovered, each with a length of 145 steps, a CI of 0.40, an HI of 0.60, an RI of 0.60, and an RC of 0.24 (Fig. 14). The tree topology was assessed using a Bremer decay index approach, and resolution was lost after the addition of one step. The analysis with the artificial adult and all 30 specimens did not recover a single most mature individual, but it did identify the group of mature *T. proriger* as more mature than that of *T. nepaeolicus*; a second analysis, which only included the nine individuals that were recovered as sister to the artificial adult in the analysis with all 30 specimens (KUVP 1032, KUVP 50090, USNM 8898, FFHM 1997-10, FMNH P15144, ROM 7906, AMNH FARB 221, FHSM VP-3, and KUVP 5033) identified FHSM VP-3 and KUVP 5033 as more mature than the others, and a comparison of the number of growth changes identified KUVP 5033 as the most mature individual.

Most of the specimens recovered by this analysis as relatively immature (stages 1–8) are *T. proriger* and are individuals that were also recovered as relatively immature (i.e., in the lower two thirds of the ontogram) in the individual analysis (Fig. 5). All but two

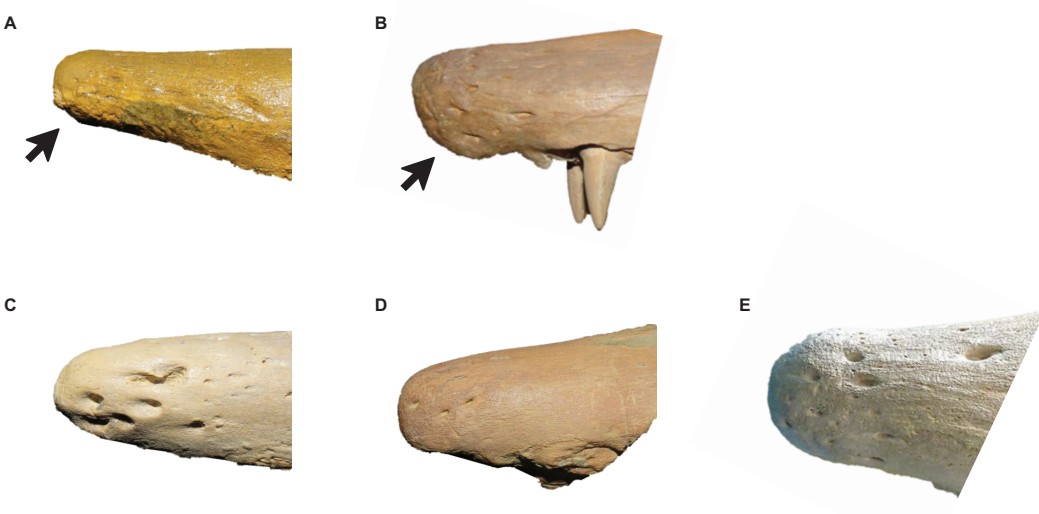

**Figure 7 Variation in *Tylosaurus* premaxillae.** Intraspecific variation of *Tylosaurus* premaxilla rostrum shape (A and B) and foramina size (C–E). In relatively immature individuals, the premaxillary rostrum is acute in lateral view (A; *T. proriger* AMNH FARB 4909) and the foramina are large (C; *T. nepaeolicus* FHSM VP-14840), whereas in mature individuals, the rostrum is rounded and distinctly knobbed (B; *T. proriger* FMNH P15144) and the foramina are either small (D; *T. nepaeolicus* FHSM VP-7262) or both small and large (E; *T. proriger* FHSM VP-3). The photographs of FMNH P15144 and FHSM VP-14040 have been inverted to face left; FHSM VP-14840 was originally identified as *T. kansasensis*; specimen photographs are not to scale.

*T. kansasensis* are recovered at growth stages 8 and 9, and all but one specimen referred to *T. nepaeolicus* are recovered at stage 10. Finally, the most mature individuals (stages 11–13) are all large (i.e., TSL greater than 1,000 mm and QH greater than 150 mm) *T. proriger*, and all but three were recovered as relatively mature (i.e., in the upper third of the ontogram) in the individual analysis (Fig. 5). Optimized synontomorphies that support each growth stage are listed in Table 5, and character states that were unambiguously optimized as individual variation are listed in Table S4. The following growth stages are recovered based on the unambiguously optimized synontomorphies that support each node on the ontogram:

**Stage 1.** This stage is recovered as sister to the other specimens (exemplar: *Tylosaurus* sp. neonate FHSM VP-14845).

**Stage 2.** The quadrate tympanic ala is thick (Fig. 6) (exemplar: *T. kansasensis* FHSM VP-9350).

**Stage 3.** The QH is between 50 and 99 mm, the quadrate infrastapedial process is present (Fig. 6), the quadrate ala rim is defined, and the quadrate mandibular condyle is completely ossified (exemplars: *T. proriger* FMNH UR902 and *T. proriger* AMNH FARB 1592).

**Stage 4.** The quadrate suprastapedial process is intermediate in length (exemplars: *T. nepaeolicus* holotype AMNH FARB 1565 and *T. proriger* RMM 5610).

**Stage 5.** The quadrate mandibular condyle is rounded (Fig. 6) (exemplars: *T. proriger* KUVP 66129).

**Stage 6.** The premaxillary rostrum is ≥5% TSL, the quadrate infrastapedial process is rounded (Fig. 6), QH is ≥13% TSL, the parietal nuchal fossa is present, and the distance between the first and sixth dentary teeth is ≤23% TSL (exemplar: *T. proriger* AMNH FARB 4909).

**Stage 7.** The foramina on the premaxillary rostrum are large (Fig. 7), the frontal-parietal suture flanges are small (Fig. 9), the jugal posteroventral process is present, and the dentary length is between 60% and 56% lower jaw length (exemplar: *T. proriger* KUVP 1033).

**Stage 8.** The parietal posterior pegs are present and small and the pterygoid ectopterygoid process is thick (Fig. 15) (exemplars: *T. proriger* KUVP 28705, *T. kansasensis* FGM V-43, *T. kansasensis* holotype FHSM VP-2295, *T. kansasensis* FHSM VP-15632, and *T. kansasensis* FHSM VP-78). At this stage, all four *T. kansasensis* share a reversal to a premaxillary rostrum that is less than 5% TSL and FHSM VP-2295, FHSM VP-15632, and FHSM VP-78 share a reversal to a quadrate infrastapedial process that is absent.

**Stage 9.** This stage is diagnosed by a reversal to frontal-parietal suture flanges that are large and a dentary length that is ≤55% lower jaw length (exemplars: *T. kansasensis* FHSM VP-15631 and *T. kansasensis* FHSM VP-2495).

**Stage 10.** The premaxillary rostrum is distinctly knobbed (Fig. 7), the frontal posterolateral processes are thick (Fig. 10), there is a reversal to parietal posterior pegs that are absent, and the coronoid anterolateral notch is present and shallow (exemplars: *T. nepaeolicus* YPM 3974, *T. nepaeolicus* AMNH FARB 124/134, *T. nepaeolicus* FHSM VP-2209, *T. nepaeolicus* FHSM VP-7262, and *T. kansasensis* FMNH PR2103). At this stage, the exemplar specimens share a quadrate infrastapedial process that is subtle and pointed (Fig. 6), parietal lateral borders that are straight (Fig. 16), and 14 dentary teeth.

**Stage 11.** The TSL is between 1,000 and 1,499 mm and the QH is between 150 and 199 mm (exemplars: *T. proriger* KUVP 1032, *T. proriger* KUVP 50090, *T. proriger* USNM 8898, *T. proriger* FFHM 1997-10, *T. proriger* FMNH P15144, *T. proriger* ROM 7906, and *T. proriger* AMNH FARB 221). At this stage, the exemplar specimens share a premaxilla–maxilla suture that is m-shaped (Fig. 8), and the relatively mature individuals (as recovered by the individual analysis (Fig. 5); FMNH P15144, ROM 7906, and AMNH FARB 221) share a reversal to a slender dentary.

**Stage 12.** The quadrate alar concavity is deep (Fig. 6) and the coronoid posteroventral process is present and fan-like (Fig. 12) (exemplar: *T. proriger* FHSM VP-3).

**Stage 13.** This stage is diagnosed by a TSL that is ≥1,500 mm and a QH that is ≥200 mm (exemplar: *T. proriger* KUVP 5033).

## Congruence between size and maturity

When the analyses were run excluding size characters, resolution was lost but the same relative positions of specimens on the ontograms was recovered (Figs. S1D–S1I). Scatterplots of size rank (TSL and QH) and growth rank data (Tables S5–S7) that were

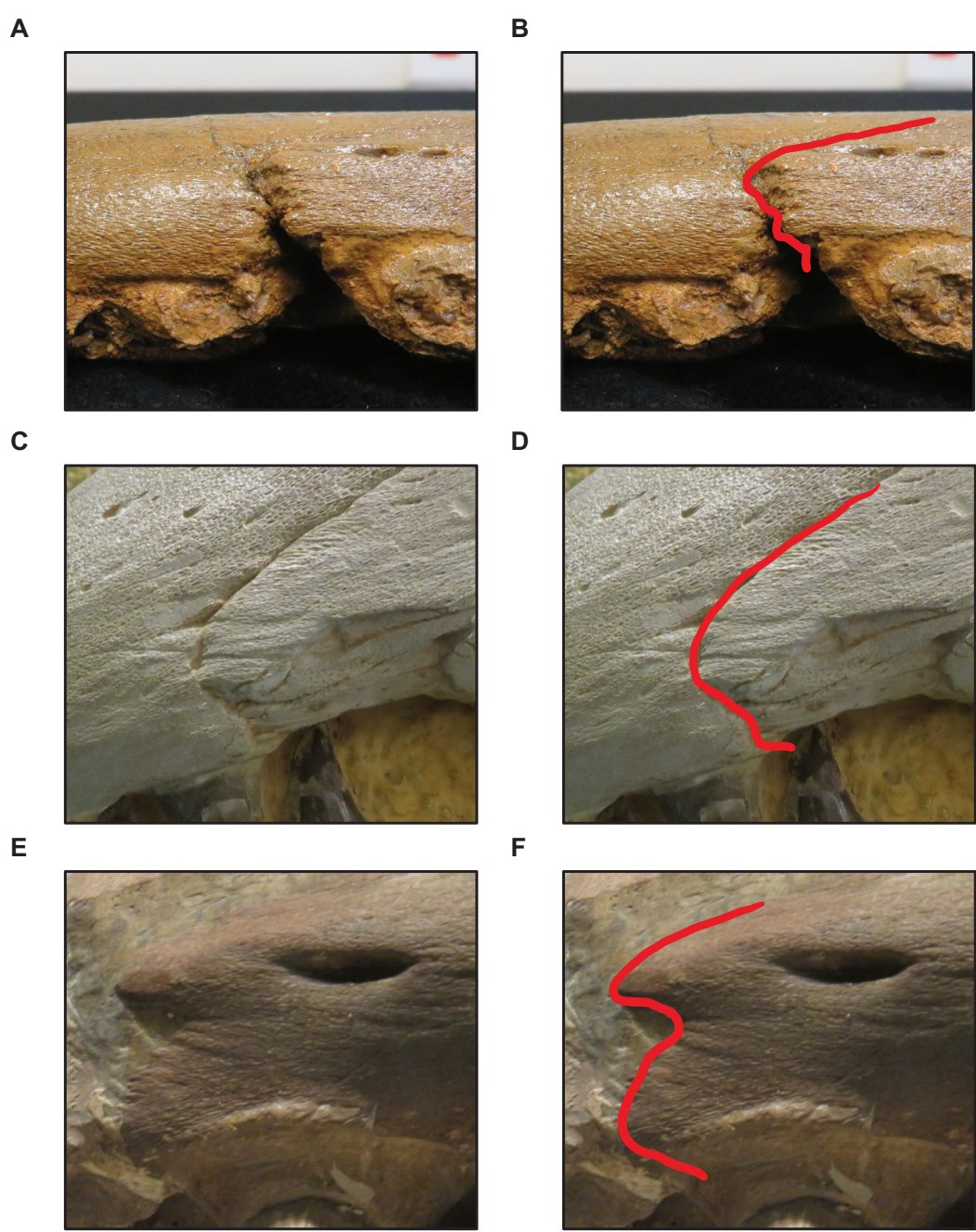

**Figure 8 Variation in premaxilla-maxilla suture shape.** (A and B) Rectangular (*T. proriger* AMNH FARB 4909). (C and D) U-shaped (*T. proriger* FHSM VP-3). (E and F) M-shaped (*T. proriger* FMNH P15144). The photograph of AMNH FARB 4909 has been inverted to face left; specimen photographs are not to scale.

used in the Spearman rank-order correlation tests are shown in Figs. 17–19. A Shapiro–Wilk test was used to determine if there was sampling bias (i.e., skewed left or right) and revealed that all the growth rank, size rank, and measurement data, except for QH growth rank data for *T. nepaeolicus*, are normally distributed (Figs. 17–19). The Spearman rank-order test found a significant correlation between growth stage and both measures of

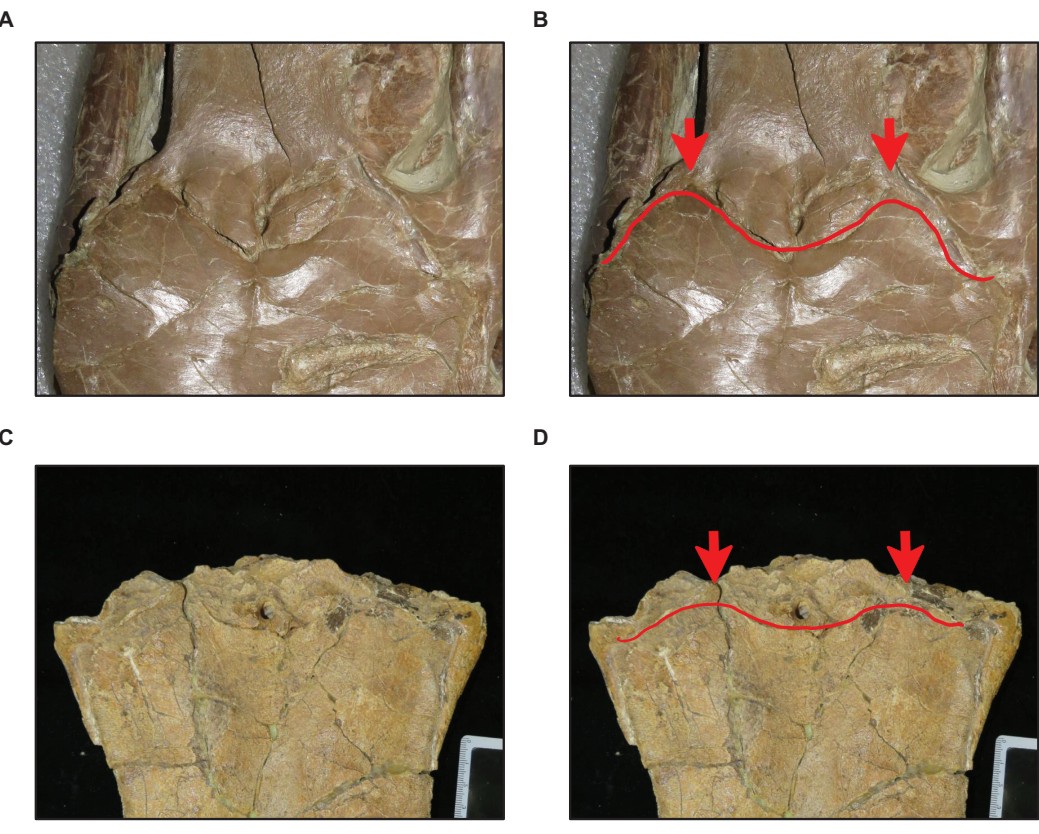

**Figure 9 Variation in frontal-parietal medial suture flange shape.** (A and B) Flanges large (*T. nepaeolicus* FHSM VP-2295). (C and D) Flanges small (*T. nepaeolicus* FHSM VP-15631). FHSM VP-2295 is the holotype of *T. kansasensis*; specimen photographs are not to scale.

size in *T. proriger* and *T. nepaeolicus*, both in the individual analyses (Figs. 17 and 18) and the analysis used to test for anagenesis (Fig. 19). All correlations between size and maturity are positive. Therefore, both TSL and QH and maturity usually covary in *Tylosaurus*.

## DISCUSSION

### Growth series of *T. proriger*

The growth series of *T. proriger* has two bifurcations, at stages nine and 14 (Fig. 5). The bifurcation at stage 14, in which three specimens share a distance between the first and sixth maxillary teeth that is ≥25% TSL and a reversal to a QH between 100 and 149 mm, does not meet the criteria of *Frederickson & Tumarkin-Deratzian (2014)* for sexual dimorphism. The group of specimens at stage nine share a distance between the first and sixth dentary teeth that is ≤23% TSL and ≤35% dentary length, which develop independently at stages 13 and 15, respectively; however, none of the growth characters separating the specimens at stage nine from those at stages ten through 17 are obviously correlated with any kind of sexual display (e.g., thickening of the quadrate suprastapedial and frontal posterolateral processes, presence of dentary predental dorsal

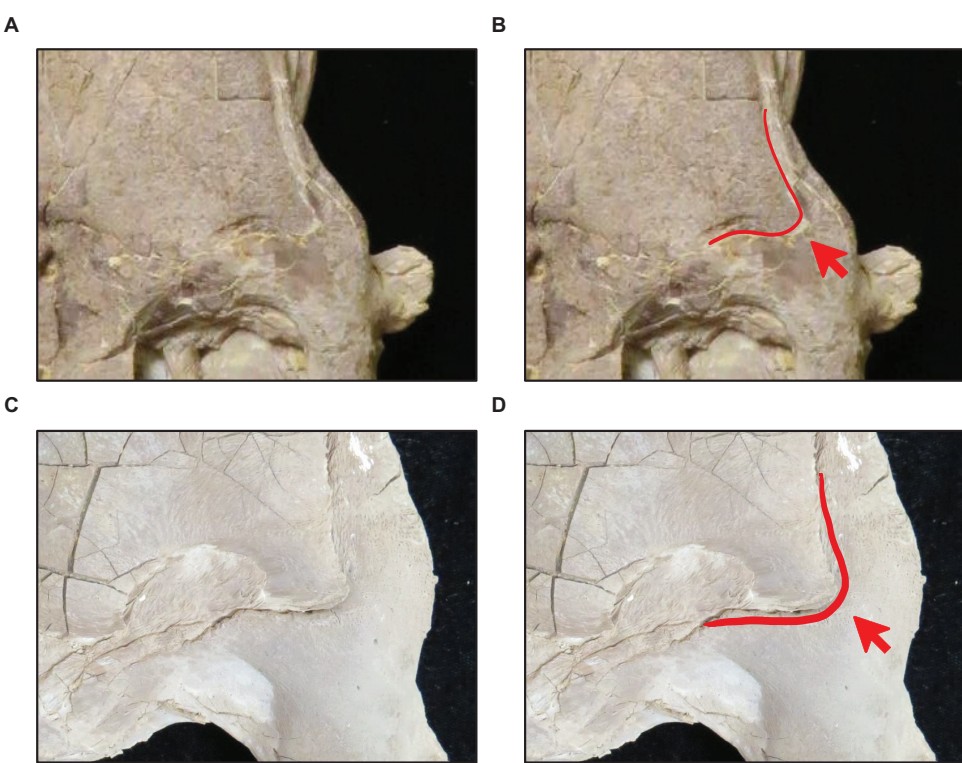

**Figure 10 Variation in frontal posterolateral process shape.** (A and B) Slender (*T. proriger* KUVP 28705). (C and D) Robust (*T. proriger* KUVP 65636). Specimen photographs are not to scale.

ridge, and knobbed premaxillary rostrum). If, however, these characters are correlated with being larger, it is possible that *T. proriger* was sexually dimorphic with respect to size–the TSL of the specimens at stage nine range from 610 mm to 813 mm (average: 712 mm), whereas the TSL of specimens from stage 10 to stage 17 are generally larger, ranging from 585 mm to 1,300 mm (average: 1,032 mm).

The major growth changes of *T. proriger* are: development of processes on the premaxilla (Fig. 7), frontal (Fig. 10), jugal, pterygoid (Fig. 15), quadrate (Figs. 6 and 20), coronoid (Fig. 12), and dentary (Fig. 11); decrease in premaxillary foramina size (Fig. 7); change in shape of the premaxilla-maxilla suture (Fig. 8); ossification of the quadrate and basioccipital; enlargement of tooth crowns relative to skull size; and a progressive deepening and enlargement of the skull. The identification of RMM 5610 as a young individual in previous work is supported, but the identification (*Jiménez-Huidobro, Simões & Caldwell, 2016*; *Stewart & Mallon, 2018*; *Jiménez-Huidobro & Caldwell, 2019*) of AMNH FARB 4909 as a relatively mature individual is not (Fig. 5).

The Spearman rank-order test revealed a significant correlation between size rank and growth stage rank for both TSL ($r_{S(0.05, 18)}$ = 0.824, $p < 0.001$) and QH ($r_{S(0.05, 17)}$ = 0.897, $p < 0.001$), suggesting that both measures are reliable proxies for relative maturity in *T. proriger* (Fig. 17). This result is unexpected, given the oversampling of relatively mature individuals: apart from the *Tylosaurus* sp. neonate (FHSM VP-14845),

A

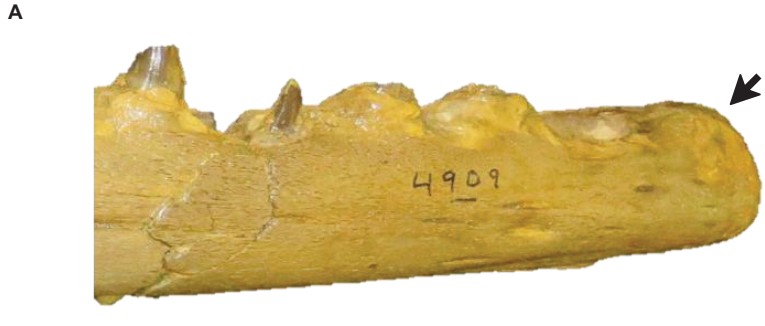

B

**Figure 11 Presence of dorsal ridge on predental process of the dentary.** (A) Absent (*T. proriger* AMNH FARB 4909). (B) Present (*T. proriger* FMNH UR820). Specimen photographs are not to scale.

this analysis only includes large (TSL greater than 500 mm) individuals. The correlation between size and maturity can be tested with the addition of significantly smaller, presumably less mature, specimens.

### Growth series of *T. kansasensis* and *T. nepaeolicus*

The ontogram does not bifurcate and so it does not show evidence for sexual dimorphism, whereas the synonymy of *T. kansasensis* with *T. nepaeolicus* (*Jiménez-Huidobro, Simões & Caldwell, 2016*) is supported (Figs. 3, 4 and 13). Most significantly, many of the diagnostic characters for *T. kansasensis* (*Everhart, 2005*) that could be identified (premaxilla foramina size, quadrate infrastapedial process, frontal midline crest, jugal posteroventral angle, quadrate ala thickness, quadrate alar concavity depth) were found to be immature characters and were also present in both *T. nepaeolicus* and *T. proriger*. Therefore, both taxa will be referred to as *T. nepaeolicus* henceforth. Although synonymy is supported, previous hypotheses of growth patterns are not, given that *T. kansasensis* specimens are interspersed among those of *T. nepaeolicus* at the terminus of the ontogram. Notably, the holotype of *T. nepaeolicus* (stage 9) is recovered as more mature than the holotype of *T. kansasensis* (stage 8) (Fig. 13); their separation is ambiguously supported by straight lateral borders of the parietal (Table 4).

The major growth trends in *T. nepaeolicus* include: enlargement of processes on the premaxilla (Fig. 7), frontal (Fig. 10), quadrate (Figs. 6 and 20), coronoid (Fig. 12), and

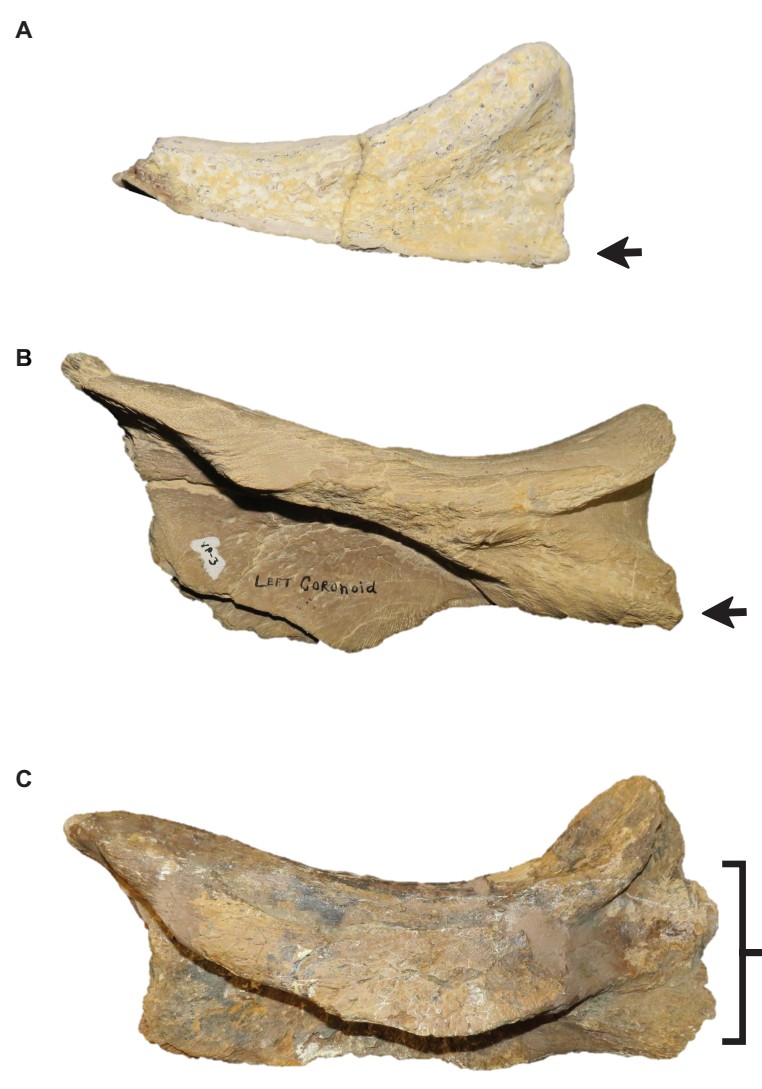

**Figure 12 Presence and shape of the coronoid posteroventral process.** (A) Absent (*Tylosaurus* sp. FHSM VP-14845). (B) Present as bump (*T. proriger* FHSM VP-3). (C) Fan-like (*T. proriger* KUVP 5033). The photograph of KUVP 5033 has been inverted to face left; specimen photographs are not to scale.

dentary (Fig. 11); change in shape of the quadrate (Figs. 6 and 20) and pterygoid (Fig. 15); changes in size of the premaxillary foramina (Fig. 7); change in shape of the premaxilla–maxilla suture (Fig. 8); ossification of the quadrate; enlargement of tooth crowns relative to skull size; and an increase in the number of dentary teeth by one.

The Spearman rank-order test revealed a significant correlation between size rank and growth stage rank for TSL ($r_{S(0.05,\ 8)} = 0.874$, $p = 0.005$) and QH ($r_{S(0.05,\ 15)} = 0.719$, $p = 0.003$), suggesting that both are reliable proxies for relative maturity in this taxon (Fig. 18). Unlike *T. proriger*, multiple specimens in this dataset are relatively small (TSL less than 500 mm, QH less than 50 mm), suggesting a better representation of immature individuals than in *T. proriger*.

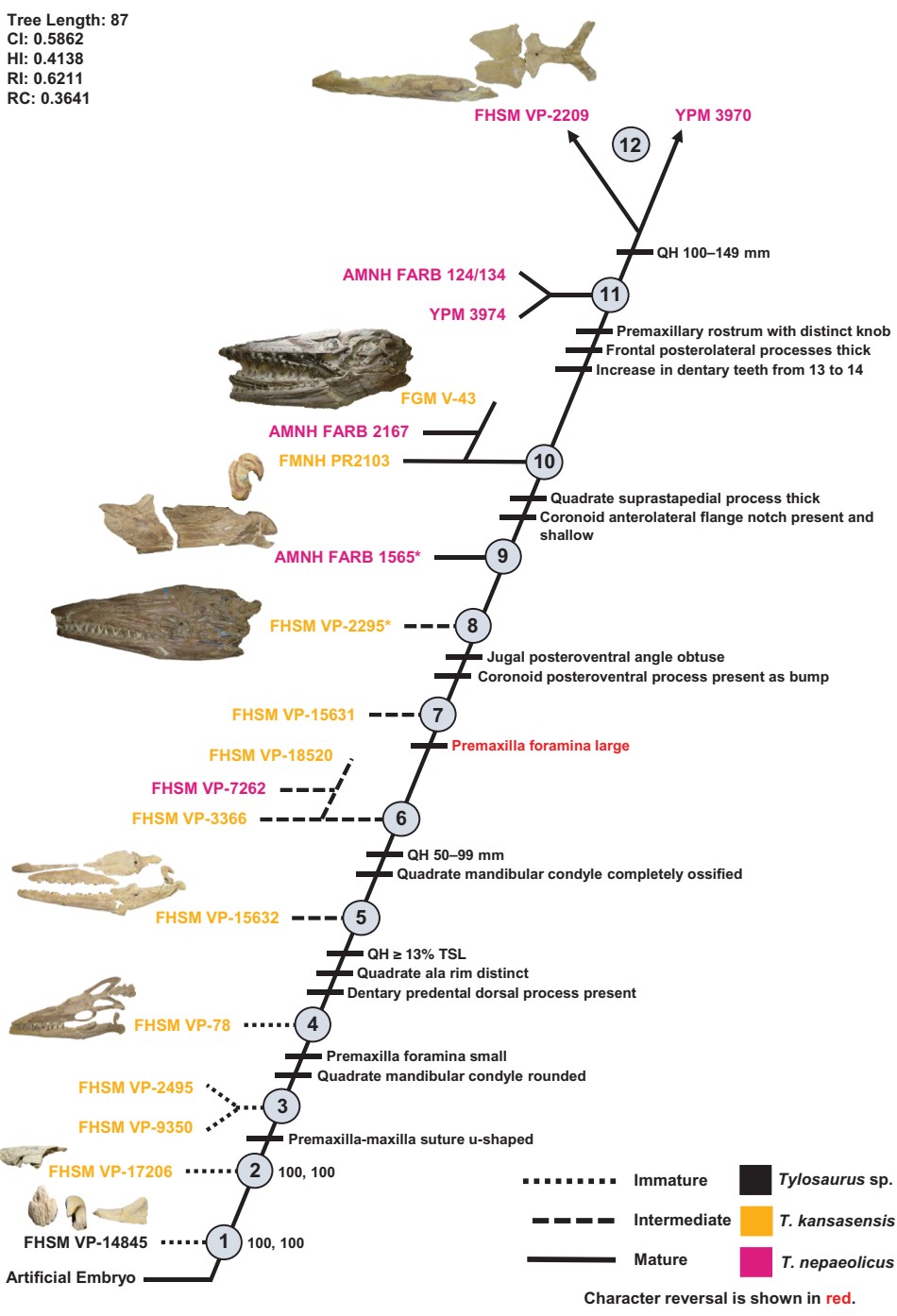

**Tree Length: 87**
**CI: 0.5862**
**HI: 0.4138**
**RI: 0.6211**
**RC: 0.3641**

**Figure 13 Ontogram of one *Tylosaurus* sp. specimen, 11 *Tylosaurus kansasensis* specimens, and seven *Tylosaurus nepaeolicus* specimens based on a quantitative cladistic analysis.** Specimens identified as *T. nepaeolicus* are shown in magenta, and specimens identified as *T. kansasensis* are shown in yellow; the type specimen of each taxon is indicated by an asterisk. The ontogram is a single tree and tree statistics are summarized in the upper left. Character states that define each growth stage are shown along the main branch, and the exemplar specimens are to the left of the main branch; the most mature individuals, identified by the analysis with an artificial adult, are indicated by arrows. The encircled numbers on the nodes are the growth stages, and the numbers to the right of them are the bootstrap and jackknife values, respectively (1000 replicates, < 50% not shown). Unambiguous character reversals are
**Figure 13** (continued)
shown in red. "Immature" specimens were recovered in the lower third of the tree, "intermediate" specimens were recovered in the middle third of the tree, and "mature" specimens were recovered in the upper third of the tree. The ontogram does not bifurcate and thus supports synonymy of *T. kansasensis* with *T. nepaeolicus* and that *T. kansasensis* represent juveniles of *T. nepaeolicus* (*Jiménez-Huidobro, Simões & Caldwell, 2016*), and does not show evidence for sexual dimorphism. Specimen photographs are not to scale; FHSM VP-14845 is a neonate only referable to *Tylosaurus* sp.; the photographs of FGM V-43, FHSM VP-2209, AMNH FARB 1565, and FHSM VP-78 have been inverted to face left.

**Table 4** Optimized synontomorphies supporting the growth stages of *Tylosaurus kansasensis/nepaeolicus*.

| Growth Stage | Unambiguous | Ambiguous |
|---|---|---|
| 1 | n/a | n/a |
| 2 | None | Quadrate tympanic ala thick**, quadrate alar concavity shallow** |
| 3 | Premaxilla-maxilla suture u-shaped | None |
| 4 | Premaxillary rostrum foramina small**, quadrate mandibular condyle rounded | Frontal-parietal suture flanges small**, jugal ascending ramus thick, pterygoid ectopterygoid process thick, basioccipital ossified |
| 5 | QH ≥13% TSL, quadrate ala rim defined, dorsal ridge of dentary predental process present | Quadrate suprastapedial process intermediate length**, parietal posterior pegs absent* |
| 6 | QH between 50 and 99 mm, quadrate mandibular condyle ossified | TSL between 400 and 800 mm, quadrate infrastapedial process present**, **frontal-parietal suture flanges large*** |
| 7 | **Premaxillary rostrum foramina large*** | Dentary deep |
| 8 | Jugal posteroventral angle obtuse**, coronoid posteroventral process present as bump* | None |
| 9 | None | Parietal lateral borders straight** |
| 10 | Quadrate suprastapedial process thick, coronoid anterolateral notch present and shallow | None |
| 11 | Premaxillary rostrum distinctly knobbed, frontal posterolateral processes thick, increase in dentary teeth* (13 to 14) | Frontal dorsal crest absent** |
| 12 | QH between 150 and 199 mm | TSL between 800 and 999 mm, premaxillary rostrum ≥5% TSL, **premaxilla-maxilla suture rectangular** |

**Note:**
Reversals are bold, phylogenetic characters are indicated by an asterisk, and characters that are purportedly diagnostic of *T. kansasensis* or *T. nepaeolicus* are indicated by two asterisks.

## Paedomorphy in *T. proriger*

Paedomorphy is the truncation of development in a descendent taxon relative to an ancestral taxon (*Reilly, Wiley & Meinhardt, 1997*). In the analysis including all three taxa, the lateral borders of the parietal table (Fig. 16) are straight in the relatively mature specimens of *T. nepaeolicus* (YPM 3974, AMNH FARB 124/134, FHSM VP-2209, FHSM VP-7262, and FMNH PR2103; Table S4), whereas they are distinctly convex in relatively immature *T. nepaeolicus* and all *T. proriger*. *Jiménez-Huidobro, Simões & Caldwell (2016)* suggested that the borders become straight in mature individuals due to elongation of the bone; truncation of this lengthening in *T. proriger* is consistent with the hypothesis of paedomorphy. Therefore, paedomorphy of this character in *T. proriger* is not rejected, but future work investigating this character (parietal lateral border shape)

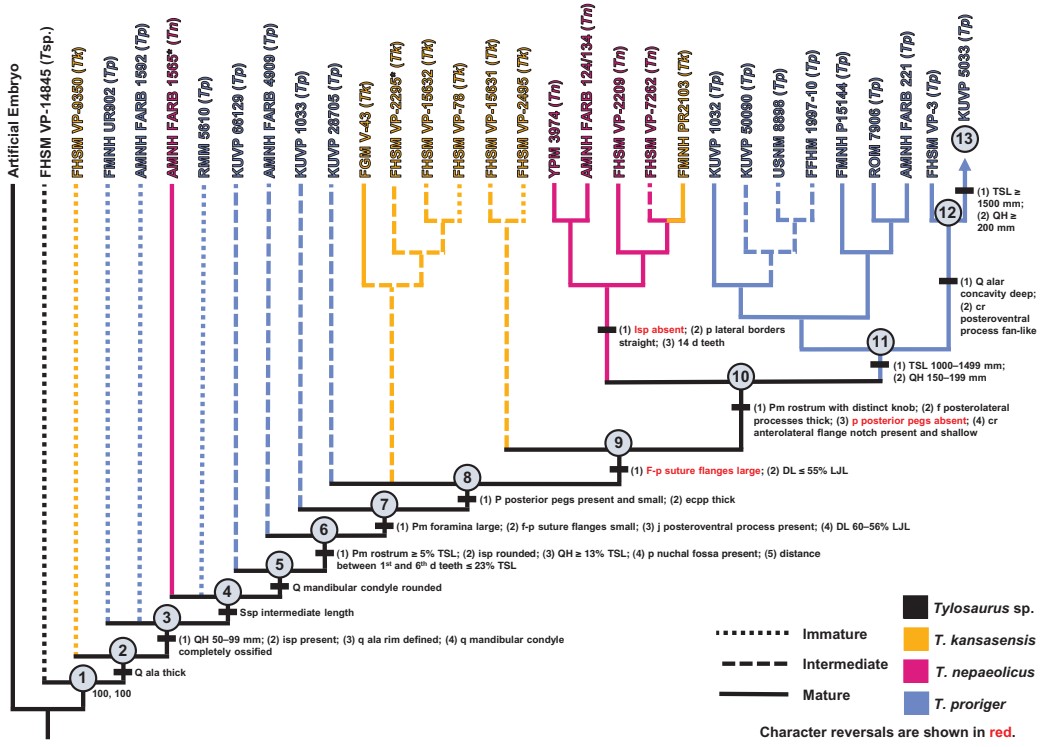

**Figure 14 Ontogram of one *Tylosaurus* sp. (Tsp.), eight *Tylosaurus kansasensis* (Tk), five *Tylosaurus nepaeolicus* (Tn), and 16 *Tylosaurus proriger* (Tp) based on a quantitative cladistic analysis.** The ontogram is based on a strict consensus of two trees, each with a length of 145 steps, a CI of 0.40, an HI of 0.60, an RI of 0.60, and an RC of 0.24. Holotypes are indicated by asterisks. Character states that diagnose each growth stage are shown along the main branch, and the exemplar specimens are to the left of the main branch; the most mature individual, identified by the analysis with an artificial adult, is indicated by an arrow. Character states that distinguish the group of mature *T. nepaeolicus* from the group of mature *T. proriger* are also shown. The encircled numbers on the nodes are the growth stages, and the numbers below and to the right of them are the bootstrap and jackknife values, respectively (1,000 replicates, <50% not shown). Unambiguous character reversals are shown in red. In the individual analyses, "immature" specimens were recovered in the lower third of the tree, "intermediate" specimens were recovered in the middle third of the tree, and "mature" specimens were recovered in the upper third of the tree. Because all "mature" *T. proriger* specimens are recovered as more mature than all *T. nepaeolicus*, the hypothesis of anagenesis in WIS *Tylosaurus* is supported; additionally, all *T. nepaeolicus* specimens (except for the holotype) are recovered as more mature than all specimens of *T. kansasensis*, supporting the hypothesis that *T. kansasensis* are juveniles (*Jiménez-Huidobro, Simões & Caldwell, 2016*). Abbreviations: **cr**, coronoid; **d**, dentary; **DL**, dentary length; **eccp**, ectopterygoid process of the pterygoid; **f**, frontal; **isp**, infrastapedial process of the quadrate; **LJL**, lower jaw length; **p**, parietal; **pm**, premaxilla; **q**, quadrate; **ssp**, suprastapedial process of the quadrate.           

in other taxa (e.g., an outgroup to *Tylosaurus*) is necessary to thoroughly test this hypothesis.

Absence of the frontal dorsal midline crest was recovered as ambiguously diagnostic of stage 11 in the ontogram of *T. nepaeolicus* (Table 4), and both specimens in which the crest is absent (AMNH FARB 124/134 and YPM 3974; Data S2) are recovered as relatively mature individuals (Fig. 13). However, given that the crest is only absent in two *T. nepaeolicus* specimens out of the 35 that were scored, more data are necessary to

**Table 5 Optimized synontomorphies supporting the growth stages of the analysis including all three taxa.**

| Growth Stage | Unambiguous | Ambiguous |
|---|---|---|
| 1 | n/a | n/a |
| 2 | Quadrate tympanic ala thick** | TSL between 400 and 800 mm, quadrate alar concavity shallow** |
| 3 | QH between 50 and 99 mm, quadrate infrastapedial process present**, quadrate ala rim defined, quadrate mandibular condyle ossified | Premaxillary rostrum foramina small**, coronoid posteroventral process present as bump* |
| 4 | Quadrate suprastapedial process intermediate length** | Basioccipital ossified |
| 5 | Quadrate mandibular condyle rounded | Premaxilla-maxilla suture m-shaped, parietal foramen bordering or invading frontal* |
| 6 | Premaxillary rostrum ≥5% TSL, quadrate infrastapedial process rounded, QH ≥ 13% TSL, parietal nuchal fossa present, distance between 1st and 6th dentary teeth ≤ 3% TSL | None |
| 7 | **Premaxillary rostrum foramina large**\*\*, frontal-parietal suture flanges small**, jugal posteroventral process present*, dentary length between 60% and 56% lower jaw length | Parietal foramen close to frontal-parietal suture* |
| 8 | Parietal posterior pegs present and small*, pterygoid ectopterygoid process thick | Premaxilla-maxilla suture u-shaped, dorsal ridge of dentary predental process present |
| 9 | **Frontal-parietal suture flanges large**\*\*, dentary length ≤55% lower jaw length | Quadrate suprastapedial process thick |
| 10 | Premaxillary rostrum distinctly knobbed, frontal posterolateral processes thick*, **parietal posterior pegs absent**\*, coronoid anterolateral notch present and shallow | None |
| 11 | TSL between 1,000 and 1,499 mm, QH between 150 and 199 mm | **Premaxillary rostrum foramina small**\*\*, premaxilla-maxilla suture terminates at or posterior to 4th maxillary tooth** |
| 12 | **Quadrate alar concavity deep**\*\*, coronoid posteroventral process present and fan-like | Frontal kite-shaped |
| 13 | TSL ≥1,400 mm, QH ≥200 mm | None |

**Note:**
Reversals are bold, phylogenetic characters are indicated by an asterisk, and characters that are purportedly diagnostic of *T. proriger*, *T. kansasensis*, or *T. nepaeolicus* are indicated by two asterisks.

test the hypothesis of paedomorphy of this character in *T. proriger*. If the addition of more characters and specimens of *T. nepaeolicus* recovers absence of the crest as an unambiguously mature character, then the addition of basal mosasaurs, such as the mosasauroid *Aigialosaurus*, as well as other derived russelosaurine taxa (e.g., *Plioplatecarpus*) will help to trace the evolution of frontal crest development across the clade for a more rigorous test of the hypothesis of paedomorphy of this character in *T. proriger*.

## Anagenesis in *T. nepaeolicus* and *T. proriger*

The ontogram recovered by the analysis of both species supports the hypothesis of anagenesis in the clade (Fig. 14). The least mature individuals in the ontogram are nearly all relatively immature *T. proriger*, the specimens of intermediate maturity are relatively mature *T. nepaeolicus*, and the most mature individuals are large, relatively mature *T. proriger*. Furthermore, the placement of all but one *T. kansasensis* as less mature than

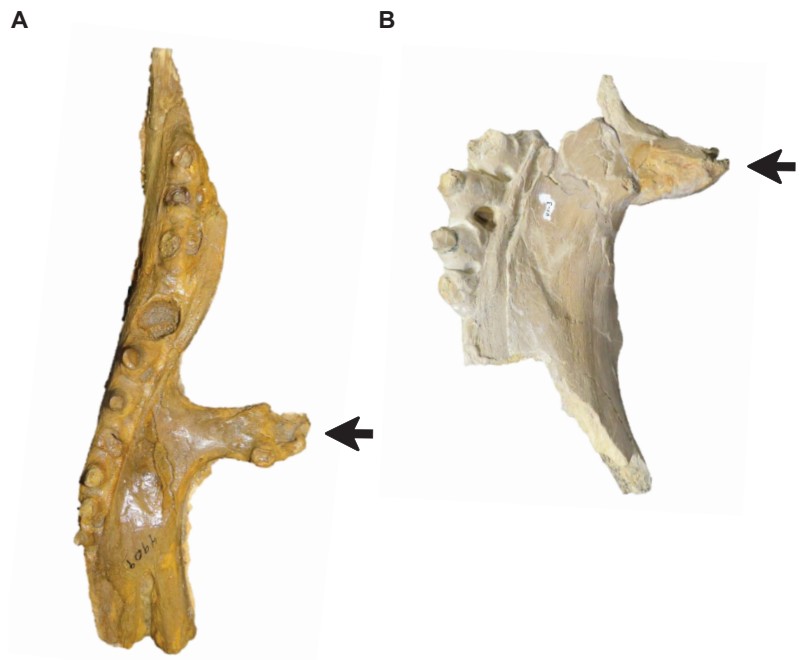

**Figure 15 Variation in pterygoid ectopterygoid process shape.** (A) Slender (*T. proriger* AMNH FARB 4909). (B) Wide and flat (*T. proriger* FHSM VP-3). Specimen photographs are not to scale.

*T. nepaeolicus* and among immature *T. proriger* is consistent with the hypothesis that *T. kansasensis* are immature *T. nepaeolicus*.

Several growth changes recovered in this analysis were also recovered in the individual analyses: thickening of the quadrate ala, quadrate mandibular condyle ossifies and becomes rounded, QH increases relative to TSL, premaxilla rostrum foramina size changes, and the frontal posterolateral processes thicken. Finally, the knobbed premaxillary rostrum (Fig. 7) after which the genus is named develops relatively late in ontogeny (at stage 11 in both individual analyses, and stage 10 in the analysis with all three taxa); therefore, not only is this character unique to *Tylosaurus*, but to the late stages of its growth, and so it is possible that young animals lacking this feature may be misidentified.

Anagenesis in WIS *Tylosaurus* was driven by peramorphosis (acceleration and/or extension of growth) in the following characters: skull size (TSL) and depth (QH) (Fig. 19), premaxillary rostrum length (greater than 5% TSL does not occur until relatively late in ontogeny in *T. nepaeolicus*, whereas it is present in immature *T. proriger*), overall quadrate shape (Figs. 6 and 20; the quadrates of the most mature *T. nepaeolicus*, e.g., AMNH FARB 124/134, are morphologically most similar to immature *T. proriger*, e.g., FMNH UR902), quadrate suprastapedial process thickness, and coronoid posteroventral process shape (from single bump to fan-like; Fig. 12). One character, lateral borders of the parietal table (Fig. 16), appears to be paedomorphic in *T. proriger*, given that it is distinctly convex in all *T. proriger* specimens as well as immature *T. nepaeolicus*,

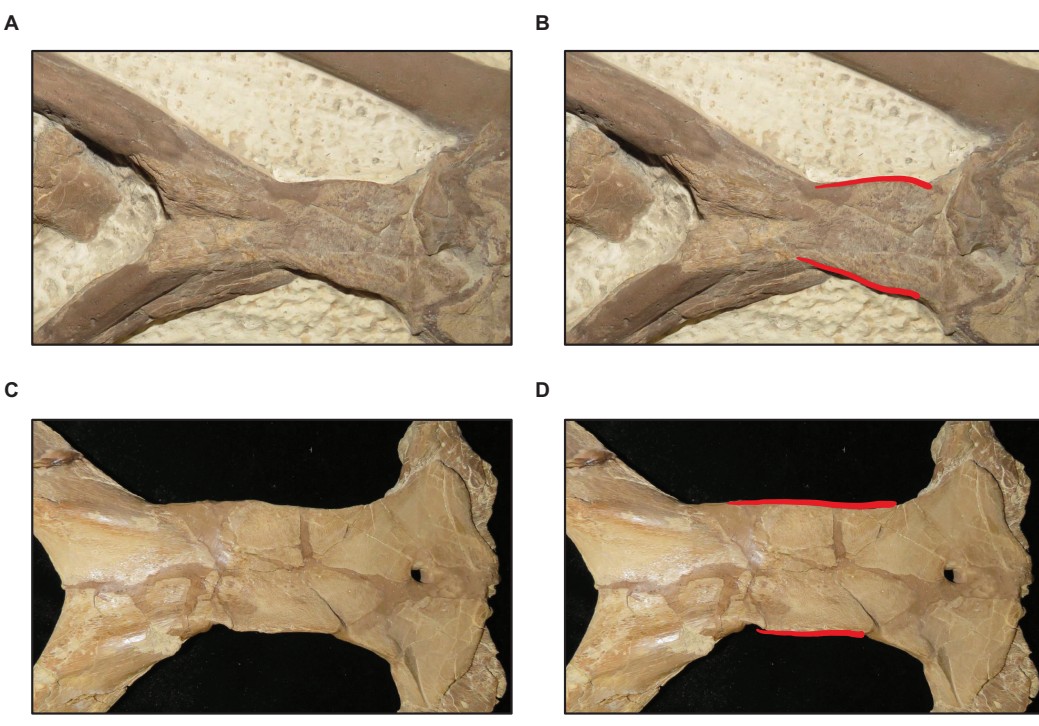

**Figure 16 Variation in parietal lateral border shape.** (A and B) Convex (*T. nepaeolicus* FHSM VP-78). (C and D) Slightly convex to straight (*T. nepaeolicus* FHSM VP-2209). FHSM VP-78 was previously identified as *T. kansasensis*; specimen photographs are not to scale.

and nearly straight in relatively mature *T. nepaeolicus*. The hypothesis of anagenesis in North American *Tylosaurus* can be further tested by recovering growth series for *T. saskatchewanensis* and *T. pembinensis*, which lived after *T. proriger* during the middle Campanian (*Jiménez-Huidobro & Caldwell, 2019*).

### Revised diagnoses of *T. nepaeolicus* and *T. proriger*

Based on the growth patterns uncovered by this work (Figs. 5, 14 and 20), the following revisions to the diagnoses of *T. proriger* by *Jiménez-Huidobro & Caldwell (2019)* are proposed: (1) premaxilla–maxilla suture ends posterior to the fourth maxillary tooth; (2) the overall shape of the quadrate is columnar and distinctly taller than wide throughout ontogeny; (3) quadrate infrastapedial process is well-developed and is subtle and pointed in immature individuals and distinct, broad, and semicircular in mature individuals; (4) quadrate tympanic ala is thin, wide, and flat throughout ontogeny; (5) lateral borders of parietal table distinctly convex; (6) 13 maxillary teeth; (7) 13 dentary teeth; and (8) ten pterygoid teeth.

Based on the growth patterns uncovered by this work (Figs. 13, 14 and 20), the following revisions to the diagnoses of *T. nepaeolicus* by *Jiménez-Huidobro & Caldwell (2019)* are proposed: (1) premaxilla–maxilla suture ends posteriorly above midpoint between third and fourth maxillary teeth; (2) frontal dorsal midline crest generally present except in some relatively mature individuals; (3) lateral borders of parietal table convex in immature individuals and slightly convex to straight in mature individuals; (4) the overall shape of the quadrate is

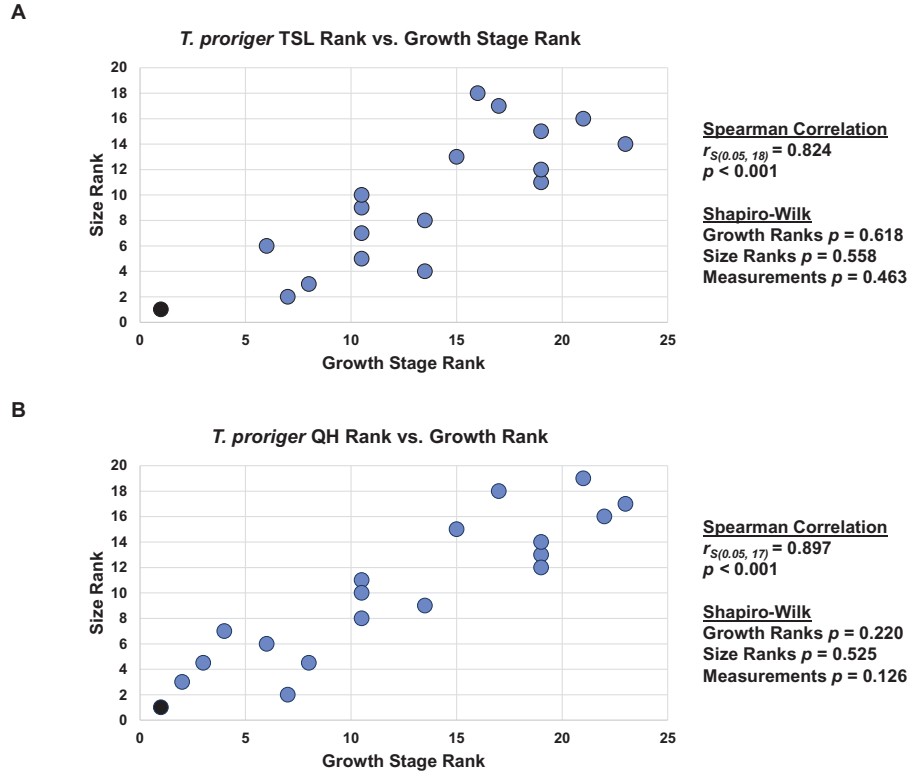

**Figure 17 Size and maturity are positively correlated in *Tylosaurus proriger*.** (A) Scatterplot and statistics for TSL data. (B) Scatterplot and statistics for QH data. The growth stages and size data for TSL and QH of each *T. proriger* (blue) and *Tylosaurus* sp. (black) specimen included in the growth series (for which measurements were available) were converted into ranks and plotted. Congruence between size rank and growth stage rank was tested with Spearman rank-order correlation. Both TSL and QH have a significant positive correlation with growth stage in this species. Shapiro–Wilk tests determined that growth rank, size rank, and raw measurement data are normally distributed.

semicircular and hook-like in immature individuals, and relatively more dorsoventrally elongate in mature individuals; (5) quadrate infrastapedial process absent in immature individuals and present but poorly developed in mature individuals; (6) 12–13 maxillary teeth; and (7) eight to ten pterygoid teeth, possibly 11 or more in immature individuals.

## Sexual dimorphism

The growth series did not recover evidence of skeletal sexual dimorphism in either species. This does not necessarily mean that *Tylosaurus* was not sexually dimorphic, only that the characters in this analysis are not dimorphic or the sample size (i.e., number of specimens) is too low for a clear pattern to be recovered (*Hone et al., 2020*). The hypothesis that the premaxillary predental rostrum is an ontogenetic, but not sexual, characteristic in *Tylosaurus* (*Konishi, Jiménez-Huidobro & Caldwell, 2018*) was not rejected. These results are consistent with the absence of evidence for sexual dimorphism in any mosasaur, which itself is somewhat surprising given the frequency of sexual dimorphism in extant squamates (*Schwarzkopf, 2005*; *Aplin, Fitch & King, 2006*;

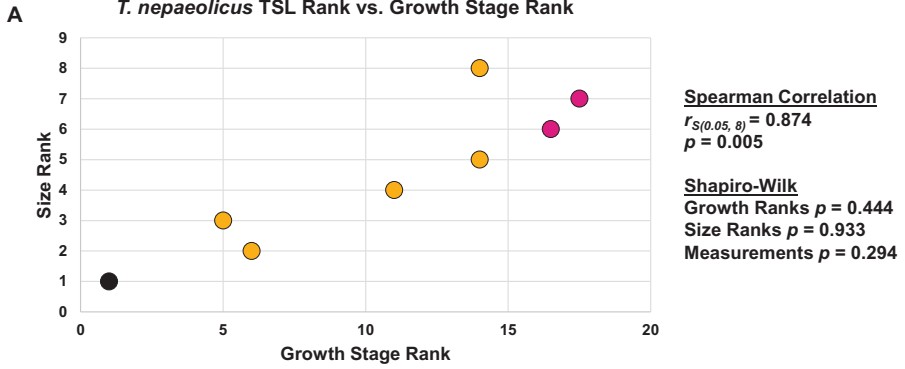

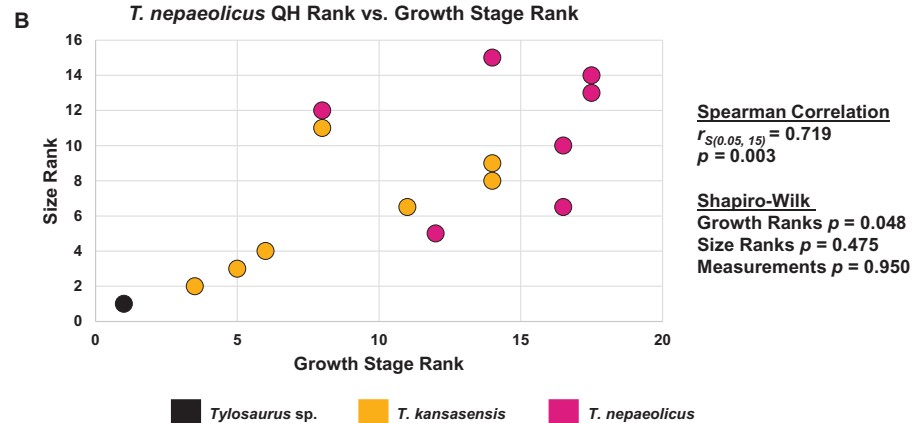

**Figure 18 Size and maturity are positively correlated in *Tylosaurus nepaeolicus*.** (A) Scatterplot and statistics for TSL data. (B) Scatterplot and statistics for QH data. The growth stages and size data for TSL and QH of each *T. nepaeolicus* (pink), *T. kansasensis* (yellow), and *Tylosaurus* sp. (black) specimen included in the growth series (for which measurements were available) were converted into ranks and plotted. Congruence between size rank and growth stage rank was tested with Spearman rank-order correlation. Both TSL and QH have a significant positive correlation with growth stage in this taxon. Shapiro–Wilk tests determined that TSL (but not QH) growth rank, size rank, and raw measurement data are normally distributed.

*Openshaw & Keogh, 2014*), including ocean-going species such as sea snakes and marine iguanas (*Wikelski & Trillmich, 1997*; *Shine et al., 2002*). Excluding size, examples of sexually dimorphic characters in extant squamates include head width, trunk length (i.e., number of presacral vertebrae), and limb length (*Schwarzkopf, 2005*). The absence of morphological sexual dimorphism in mosasaurs will be tested further with the addition of more growth characters—especially from the postcranial skeleton—as well as more specimens and taxa.

It is also possible that sexual dimorphism in mosasaurs could be present in histological data. In 2010, *Frynta et al. (2010)* found that adult male monitor lizards (*Varanus indicus*) are larger than females because the period of rapid growth is extended; if this is also the case in mosasaurs, these differences in growth rates are seen in histological analyses of limb bones (*Pellegrini, 2007*; *Green, 2018*). Another instance of sexual dimorphism in extant monitor lizards is bone density: in males, density tends to increase over time,

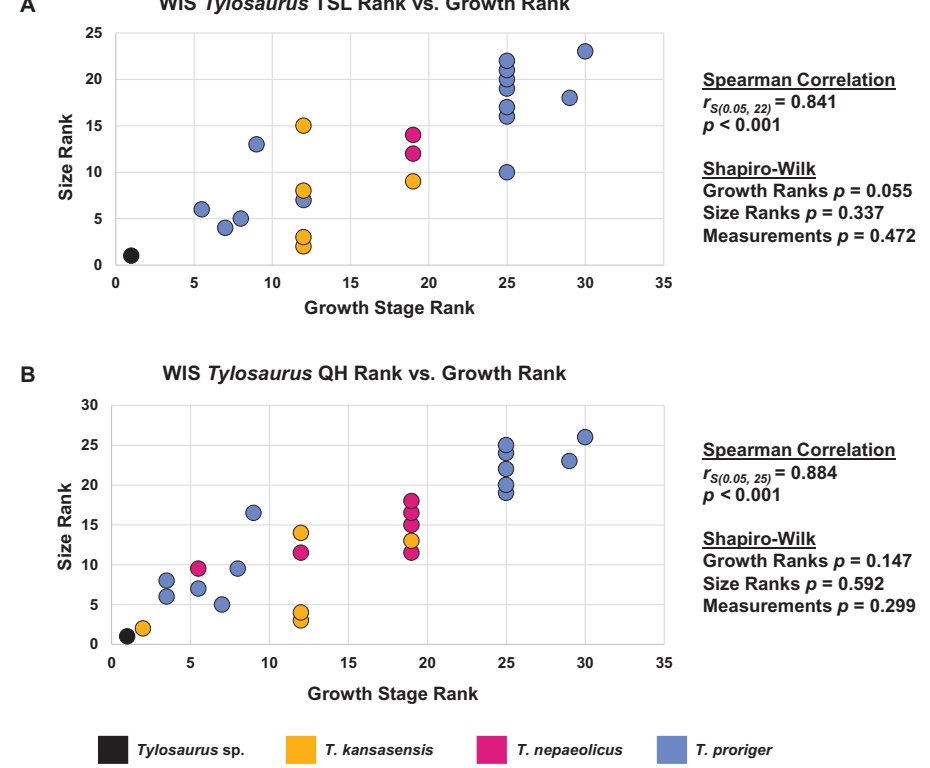

**Figure 19 Size and maturity are positively correlated in WIS *Tylosaurus* species.** (A) Scatterplot and statistics for TSL data. (B) Scatterplot and statistics for QH data. The growth stages and size data for TSL and QH of each specimen (for which measurements were available) included in the growth series including all three *Tylosaurus* taxa were converted into ranks and plotted. Congruence between size rank and growth stage rank was tested with Spearman rank-order correlation. Both TSL and QH have a significant positive correlation with growth stage. Shapiro–Wilk tests determined that growth rank, size rank, and raw measurement data are normally distributed.

whereas in females it decreases (*De Buffrénil & Francillon-Vieillot, 2001*). However, because this decrease in females is caused by skeletal calcium being used to produce eggshells, this is unlikely to be seen in mosasaurs, which gave live birth (*Caldwell & Lee, 2001*; *Field et al., 2015*).

## Cladistic analysis of growth as a method to test taxon validity

Besides traditional comparison of morphological characters, no thorough, objective tests of taxon validity using growth data have been attempted for any mosasaur taxon to date. By recovering synontomorphies and identifying instances of individual variation in multiple taxa, cladistic analysis of growth provides a robust and independent test of taxon validity and the characters that purportedly diagnose them. Taxon validity is a major problem in mosasaurs for multiple reasons, including insufficient descriptions and later loss or destruction of type specimens, paraphyly of genera, poor stratigraphic data, incompleteness of specimens, and past researchers' desire to name as many species as possible (*Lively, 2018*). This problem is only made worse by a lack of growth studies that

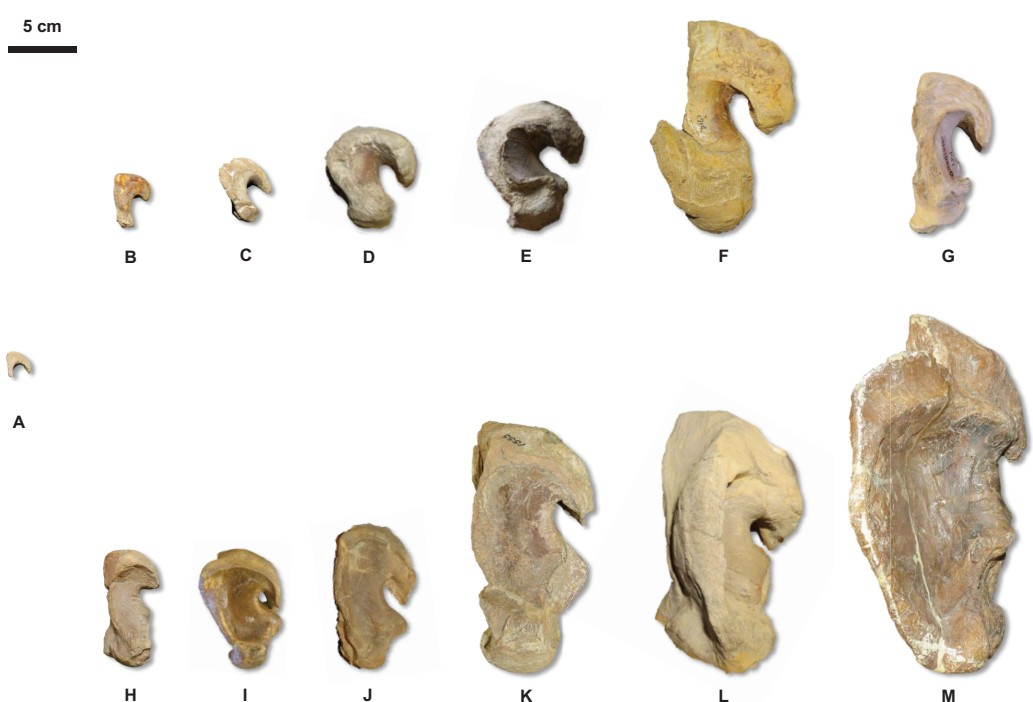

**Figure 20 Quadrate growth in WIS *Tylosaurus*.** Growth series of *Tylosaurus* sp. (A), *T. nepaeolicus* (B–G) and *T. proriger* (H–M) quadrates. (A) FHSM VP-14845. (B) FHSM VP-9350. (C) FHSM VP-15632. (D) FHSM VP-2295. (E) FGM V-43. (F) AMNH FARB 2167. (G) AMNH FARB 124/134. (H) FMNH UR902. (I) AMNH FARB 4909. (J) KUVP 1033. (K) AMNH FARB 1555. (L) FHSM VP-3. (M) KUVP 5033. Scale bar is 5 cm. FHSM VP-14845 is ventrally incomplete; the photographs of FMNH UR902, FHSM VP-15632, FGM V-43, and AMNH FARB 124/134 have been inverted to face left.

include morphological data, which could be contributing to purported differences between taxa, and a general deficiency of recent hypothesis-driven work.

For example, *Mosasaurus* is a particularly problematic group with respect to taxonomy and for which this approach could prove very useful in determining which species are valid and which are not. *Mulder (1999)* proposed that *M. maximus*—found along the east coast of the United States—and *M. hoffmannii*—found in western Africa, Russia, and across Europe—are a single, transatlantic taxon based on many morphological similarities. In addition to *M. maximus*, two other *Mosasaurus* taxa, *M. lemonnieri* and *M. conodon*, have been proposed to be synonymous with *M. hoffmannii* (*Russell, 1967*; *Lingham-Soliar, 1995*; *Lingham-Soliar, 2000*; *Ikejiri & Lucas, 2015*; *Street & Caldwell, 2017*); specimens of *M. lemonnieri* in particular have the potential to represent immature *M. hoffmannii*, given that the only major difference between them is that the skull of *M. lemonnieri* is generally smaller (around 500 mm—a size currently underrepresented in *M. hoffmannii*) and more slender than that of *M. hoffmannii* (*Lingham-Soliar, 2000*). By using a single cladistic analysis of growth including specimens of all *Mosasaurus* species for which synonymy has been proposed, as was done in this project for *T. kansasensis* and *T. nepaeolicus*, these hypotheses can be tested, refining our understanding of mosasaur growth as well as their actual diversity in the Late Cretaceous.

## Conserved patterns of growth in *Tylosaurus* and future works

The cladistic analyses of ontogeny identified 11 growth characters shared by both species; these characters are: (1) premaxilla rostrum becomes distinctly knobbed; (2) change in premaxillary rostrum foramina size; (3) change in premaxilla-maxilla suture shape; (4) increase in QH; (5) thickening of quadrate suprastapedial process; (6) increase in QH relative to TSL; (7) ossification of the quadrate mandibular condyle; (8) mandibular condyle of the quadrate becomes rounded; (9) thickening of frontal posterolateral processes; (10) development of a dorsal ridge on the predentary process of the dentary; and (11) growth of the coronoid posteroventral process.

These results reject previous hypotheses that variation of mosasaur quadrates is ontogenetically uninformative (*Jiménez-Huidobro, Simões & Caldwell, 2016*; *Stewart & Mallon, 2018*), where both species show unambiguous changes to the shape of the quadrate and its processes throughout growth (Figs. 6 and 20). This suggests that the quadrate— particularly the thickness of the suprastapedial process, depth and thickness of the tympanic ala, and the presence and shape of the infrastapedial processes—should not be used to diagnose mosasaur taxa without an assessment of maturity. These results are not surprising, given that growth variation is seen in the quadrates of extant squamates (*Paluh, Olgun & Bauer, 2018*). Because the shape of the quadrate in squamates is directly related to hearing ability and skull kinesis (*LeBlanc, Caldwell & Lindgren, 2013*; *Paluh, Olgun & Bauer, 2018*; *Palci et al., 2019*), future work is necessary to investigate the potential for niche partitioning between mosasaur growth stages.

Although size and maturity covary in both species, there is clearly an oversampling of relatively mature individuals, where multiple individuals are recovered at the same growth stage (Figs. 5, 13 and 14). Therefore, more characters must be identified to test these low-resolution results. Several skulls in this project have associated vertebrae and limb bones; future work including histological data could be used to calibrate the growth series recovered here to chronological age and further test hypotheses of the relationship between size, maturity, and age as well as sexual dimorphism and ontogenetic niche partitioning (*Wiffen et al., 1995*; *De Buffrénil & Francillon-Vieillot, 2001*; *Pellegrini, 2007*; *Frynta et al., 2010*; *Houssaye & Tafforeau, 2012*; *Green, 2018*).

The size of the foramina on the premaxillary rostrum (Fig. 7) were recovered as ontogenetically variable. Recent work on the tylosaurine *Taniwhasaurus antarcticus* has found internal branching structures, hypothesized to be part of the neurovascular system, associated with these foramina (*Álvarez-Herrera, Agnolin & Novas, 2020*). Future work can investigate whether these internal structures are also present in other mosasaurs, including *Tylosaurus*, whether they vary with growth as well, and, if so, whether these variations in size have functional implications.

No pattern of ontogenetic change in pterygoid tooth count was recovered in these analyses, however, the number of pterygoid teeth could potentially be indicators of relative maturity in mosasaurs, given that their presence and number vary ontogenetically in extant lizards (*Barahona & Barbadillo, 1998*; *Skawiński, Borczyk & Turniak, 2017*). Mosasaur pterygoid teeth have largely been ignored in the literature with respect to

both ontogeny and phylogeny, and so future studies that include them are necessary to better understand their relevance to mosasaur development and evolution, and whether intraspecific differences in the number of pterygoid teeth represent growth, sexual, or individual variation. For example, a basal russelosaurine, *Tethysaurus nopcsai*, has nearly double the number of pterygoid teeth than both species of *Tylosaurus* (*Bardet, Suberbiola & Jalil, 2003*), and a relatively immature *T. nepaeolicus* specimen (FHSM VP-15632) has more pterygoid teeth (at least 11) than more mature specimens (usually between eight and ten; Table 2).

## Synthesis of ontogeny and phylogeny

Despite many studies that have investigated mosasaur phylogeny (*Russell, 1967*; *Bell, 1997*; *Simões et al., 2017*; *Jiménez-Huidobro & Caldwell, 2019*), the evolutionary relationships within Mosasauroidea remain unclear. In order to completely investigate ancestral patterns of growth in mosasaurs, growth series for basal mosasaurs, such as *Aigialosaurus*, as well as more derived taxa spanning a greater breadth of the phylogeny (e.g., *Mosasaurus, Clidastes, Platecarpus, Prognathodon*) must be recovered. Once they are identified, ontogenetic changes that are unique to a taxon can help to recover evolutionary relationships (i.e., growth changes can be used as phylogenetic characters) (*Bhullar, 2012*); therefore, the identification of shared growth characters can provide evidence to support or reject current hypotheses of relationships between mosasaurs and their extant relatives.

For example, one growth character recovered in this project—decrease of the posteroventral angle of the jugal in *T. nepaeolicus* throughout growth—was found by *Bhullar (2012)* to be apomorphic of Varanoidea. Despite the ambiguity with respect to the position of Mosasauroidea within Squamata (*Russell, 1967*; *Carroll & DeBraga, 1992*; *Caldwell, Carroll & Kaiser, 1995*; *Lee, 1997*; *Caldwell, 1999*; *Gauthier et al., 2012*; *Reeder et al., 2015*; *Simões et al., 2017*), this character is almost certainly plesiomorphic in the common ancestor of Varanoidea and Mosasauroidea. Furthermore, the recovery of shared patterns of growth that unite mosasaurs with their extant relatives has the potential to provide a comparative point of reference for predicting the growth patterns of fossil taxa with low sample sizes (*Witmer, 1995*; *Bhullar, 2012*).

With the addition of extant relatives (e.g., monitor lizards, iguanas, and snakes), ontogenetic data can also be used to hypothesize the phylogenetic position of Mosasauroidea and identify the potential heterochronic processes that shaped the land-sea transition of mosasaur ancestors. For example, in squamates, the overall shape of an animal's quadrate is related to what type of habitat it occupies (e.g., terrestrial, aquatic, or fossorial) (*Palci et al., 2019*), and squamate quadrates change in shape throughout ontogeny (Fig. 20; *Bhullar, 2012*; *Paluh, Olgun & Bauer, 2018*). Therefore, through the comparison of patterns of growth between extant terrestrial and semi-aquatic squamates to those seen across Mosasauroidea, the changes in quadrate shape that facilitate the transition from land to sea could be traced.

Finally, comparison of growth patterns with other secondarily aquatic taxa, both extant (e.g., sirenians, pinnipeds, cetaceans, turtles) and extinct (e.g., thalattosuchians,

plesiosaurs, ichthyosaurs), is necessary to uncover the heterochronic processes that drive amniote land-sea transitions. For example, using anatomical network analysis, *Fernández et al. (2020)* found that there are two main mechanisms by which secondarily aquatic tetrapods form fins from limbs: persistence of interdigital and superficial connective tissues (seen in mosasaurs and plesiosaurs), and reintegration of the digits with the mesopodium (seen in ichthyosaurs). Additionally, *Schwab et al. (2020)* found that, in the evolution of thalattosuchians, a lineage of fully aquatic crocodylomorphs, the inner ear labyrinth became more thick and compact gradually; this is different from cetaceans, which evolved relatively small inner ear labyrinths very quickly, and suggests that the semiaquatic phase of thalattosuchian evolution lasted longer than that of cetaceans. The advantage of comparing these and other features associated with an aquatic lifestyle (e.g., shortening of long bones, nostril retraction, increase in orbit size) across lineages and in an ontogenetic context is that it can identify the heterochronic processes that drove each transition and determine whether each instance is novel or convergent with respect to fundamental developmental mechanisms.

## CONCLUSIONS

In conclusion: (1) a growth series was recovered for both species; (2) size and growth covary in *Tylosaurus*; (3) there is no evidence for skeletal sexual dimorphism in *T. proriger* or *T. nepaeolicus*; (4) synonymy of *T. kansasensis* with *T. nepaeolicus* and the hypothesis that *T. kansasensis* represent immature *T. nepaeolicus* is supported; (5) the hypothesis that the convex lateral borders of the parietal table in *T. proriger* are paedomorphic relative to *T. nepaeolicus* is not rejected, and it is unclear whether the presence of a frontal dorsal midline crest in *T. proriger* is paedomorphic relative to *T. nepaeolicus*; (6) the hypothesis that *T. nepaeolicus* and *T. proriger* are a single anagenetic lineage is supported, where speciation was driven mainly by peramorphy; (7) cranial diagnoses of *T. proriger* and *T. nepaeolicus* including ontogenetic context have been proposed; and (8) 11 shared growth changes were recovered for the genus *Tylosaurus*.

## INSTITUTIONAL ABBREVIATIONS

**AMNH**   American Museum of Natural History, New York, New York
**CMN**   Canadian Museum of Nature, Aylmer, Quebec, Canada
**FFHM**   Fick Fossil and History Museum, Oakley, Kansas
**FHSM**   Fort Hays Sternberg Museum, Fort Hays, Kansas
**FGM**   Fryxell Geology Museum, Augustana College, Rock Island, Illinois
**FMNH**   Field Museum of Natural History, Chicago, Illinois
**GSM**   Georgia Southern Museum, Statesboro, Georgia
**HMG**   Hobetsu Museum, Hokkaido, Japan
**IPB**   Goldfuss Museum im Institut für Paläontologie, Bonn, Germany
**KUVP**   University of Kansas Museum of Natural History, Lawrence, Kansas
**LACMNH**   Los Angeles County Museum, Los Angeles, California
**MCZ**   Museum of Comparative Zoology, Harvard University, Cambridge, Massachusetts

| RMM | Red Mountain Museum, Birmingham, Alabama |
| ROM | Royal Ontario Museum, Toronto, Ontario, Canada |
| TMM | Texas Memorial Museum, University of Texas, Austin, Texas |
| TMP | Royal Tyrrell Museum of Palaeontology, Drumheller, Alberta, Canada |
| USNM | United States National Museum, Washington, D.C. |
| YPM | Yale Peabody Museum, Yale University, New Haven, Connecticut |

## ACKNOWLEDGEMENTS

I thank T. Carr for mentorship, feedback on this project, and for reviewing the manuscript prior to submission. I also thank T. Burling, D. Choffnes, T. Gamble, A. Griffing, and B. Holbach for general help, support, and advice. Additionally, I thank M. Everhart (FHSM) for having specimen photos and other resources available on his website and for clarifying information about AMNH FARB 221, FGM V-43, and KUVP 5033. For access to specimens, I thank: W. Simpson, A. Stroup (FMNH); C. Mehling, M. Norell (AMNH); S. Kornreich Wolf (FGM); C. Shelburne (FHSM); and M. Sims, A. Whitaker (KU). I thank J. Mallon (CMN) and T. Konishi (University of Cincinnati) for additional information about FFHM 1997-10 and CMN 8162, respectively, and P. Jiménez-Huidobro for clarifying some of the growth characters. The program TNT was made available by the Willi Hennig Society. Finally, I thank the reviewers (J. Frederickson, A. LeBlanc, T. Konishi) and the primary editor (V. Abdala) for their comments, which greatly improved the clarity and overall quality of this work.

### Funding

Carthage College funded travel to present preliminary results of this project at both the 79th Annual Meeting of the Society of Vertebrate Paleontology in Brisbane, Queensland, Australia, and, with permission from J. Mathews, at the 22nd Annual PaleoFest at the Burpee Museum of Natural History in Rockford, IL. The funders had no role in study design, data collection and analysis, decision to publish, or preparation of the manuscript.

### Grant Disclosures

The following grant information was disclosed by the authors:
Carthage College.

### Competing Interests

The author declares that they have no competing interests.

### Author Contributions

- Amelia R. Zietlow conceived and designed the experiments, performed the experiments, analyzed the data, prepared figures and/or tables, authored or reviewed drafts of the paper, and approved the final draft.

## Data Availability

Raw data are available in the Supplemental Files.

## Supplemental Information

Supplemental information for this article can be found online at http://dx.doi.org/10.7717/peerj.10145#supplemental-information.

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
