# Peer review of "Craniofacial ontogeny in Tylosaurinae"

_PeerJ, doi:10.7717/peerj.10145_

## Round 0.1 · original submission · Major Revisions

I received three mostly positive reviews of your manuscript. However, it still needs considerable work in edition and in improving some methodological issues.

Reviewer #1 has doubts about your criteria for optimization, size characters, and codification of some characters.

Reviewer#2 is concerned about the sample used, combining specimens examined by you and others from other sources. Although I have nothing against using data from the literature or internet databases, I would like you to follow our second reviewer's suggestion to repeat the analyses considering only the data you observed.

Reviewer#3 believes that the process used to determine ontogenetic stages using osteological characters is circular.

All reviewers make a detailed edition of your manuscript. Please, take these and all other suggestions into account to improve the quality of your work.

·

Basic reporting

The paper is generally written well, but could use some editing to become less redundant and improve the readability. Specifically:

Line 36: remove "Literature Review" subheading as it is implied by an introduction.

Lines 43, 48, 55, 66, 73: instead of starting every line with "In XXXX, Author", use Author (XXXX). For example, Line 43 "In 2007, Caldwell published..." can be "Caldwell (2007) published". This will make the start of every paragraph less repetitive.

Lines 37 to 80: there is no need to cite the same paper multiple times in the same paragraph if it is the only one you are discussing.

Line 239: this paragraph is a bit difficult to follow for someone unfamiliar with size-independent ontogenetic studies. You're making two arguments here about size, (1) that size differences can be phylogenetic or ontogenetic variation and (2) size is variable and normally can't be used to reliably assess maturity beyond broad groupings. I suggest adding to this section to flesh out these separate ideas.

Line 269: you could cite Carr (2020) or Frederickson and Tumarkin-Deratzian (2014) for the definition of synontomorphy as well.

Line 270: "Kansas University" is technically the "University of Kansas". I'd use the Kansas Museum of Natural History here instead.

Line 282: there is no need for initials since you are the only author.

Line 311: those tables are results.

Line 329: put hypothetically before "will". I'll discuss this more in the next section.

Line 370: "one the other hand" should be "on the other hand".

Line 370 to 372: this paragraph is a bit confusing as it comes across like you haven't already done the analysis. I understand why you presented both, but in reality you only need to discuss the one you actually did.

Line 417: small and large are relative terms. If you can quantify them or show a figure (like in the SD), I would do that.

Line 427 (and throughout): less than or equal to is more easily shown as the symbol.

Line 690: All hypotheses can be refined with the addition of more data. A similar line to this comes up many times in the discussion. I would move it to the conclusions.

Line 703: E in Figure 5 shows another possibility for dimorphism without bifurcation.

Lines 720 to 732: this paragraph seems redundant with the previous one. Consider combining them into one section.

Line 999: Royal Tyrrell Museum of Palaeontology.

The literature cited is comprehensive. I'd suggest adding Carr, 2020 to the manuscript as it was recently published. There are multiple errors in the work cited outlined below:

"and" should be "&" in internal references (throughout)
Comma needs to be removed after journal (lines 1054, 1068, 1079, 1085, 1134, 1148)
Capitalization errors (lines: 1060 Vertebrate; 1066 Reports; 1088 Science; 1159 & 1182 PLOS ONE; 1201 Functional...)
No periods for author initials (throughout)
No "and" in multi-authored papers (throughout)

The figures are great and the first three are the first time I've seen a visual description of the predicted outcomes outlined in other cladistic ontogeny studies. I would even argue that some of the important anatomical figures from the supplementary information would be appropriate to move to the main text.

Experimental design

SD character 1 and 10: Is there a reason for the character optimization for TSL and QH? The character states appear to be random intervals (e.g., Less than 400 [399], 400 to 799 [399], 800-999 [199], 1000-1499 [499].

SD character 1 and 10: What happens if you run the analysis without the size characters? One could argue that using size characters and then a correlation for size is circular reasoning.

Line 262: unordered and equally weighted?

Line 264: and since no single juvenile specimen has a complete character coding of 0s.

Line 265: how did you handle coding for unordered, multi-state characters in the artificial adult?

Line 286: whole number to what length unit (e.g., centimeter)?

Line 303: what do you mean removed due to incomplete coding? It would be helpful to elaborate here on how you decided to remove these uninformative specimens.

Line 319: this is still hypothetical as nobody has yet shown how sexual dimorphism can alter the tree topology in cladistic studies of ontogeny. Though it is possible or even reasonable to assume it may bifurcate, I'd treat this as a working hypothesis by Frederickson and Tumarkin-Deratzian (2014) rather than a definitive test. Add some caution here (such as inserting "hypothetically" before bifurcate in line 329) and I think it is acceptable.

Line 454 (and multiple other places): What is the justification for defining stages that are not supported by synontomorphies?

Line 756: this is an interesting and clever way to test for anagenesis.

Validity of the findings

Overall, I see no serious concerns over the findings and interpretation. I have outlined two areas below where discussion could be added.

Line 680: see Hone et al., 2020 on sexual dimorphism and sexual size dimorphism. We would likely need an obvious character and/or a very large sample to distinguish. Though I agree with your interpretation for the current data set, it may still be too small to rule out.

Line 919: what does changing the size imply for the function of these neurovascular canals? Do you think this also relates to changes in niches between juveniles and adults? I am intrigued by the broader ecological implications of these findings.

Additional comments

This paper is an impressive and important contribution to our understanding of mosasaur ontogeny. I feel that this paper will be worthy of publication in PeerJ following the revisions and some elaboration outlined above.

·

Basic reporting

My major issue here is that most of the important anatomical data the readers need to see are tucked away in supplementary information. The ontograms presented in this paper are built on the author's choices of characters and character states, so they should be front and centre in these figures. Fortunately there is space for this. Figures 2–5 do not really add anything to the manuscript (they are verbally explained in the text already). This frees up "room" to detail the ontogenetic characters and show the discrete variation in bones/sutures.

I am also unclear how any of the continuous variables (e.g. quadrate height/proportional characters) were partitioned into discrete states. Sometimes it appears to be arbitrary (e.g., dividing measurements into multiples of 50mm as discrete states). Are these partitions biologically meaningful? Please clarify this in the Methods, but also consider the extensive body of literature on the topic of partitioning continuous anatomical data into discrete characters for cladistic analysis.

Experimental design

This study appears to follow the rationale from previously published papers on how to construct ontogenetic trajectories using a cladistic approach. I do have some conceptual and technical questions throughout the manuscript, but I refer the author to the attached PDF for those.

The strength of the study is in the section on Tylosaurus. While the author includes over 100 specimens for this analysis, they have personally observed less than half of those. This is still an impressive number of personally observed and photographed specimens, but the manuscript overreaches by including two additional mosasaur taxa (Mosasaurus hofmannii and Tethysaurus nopscai) for which they have not collected any of the data themselves. These two additional analyses need to be removed.

Furthermore, I would re-run the analyses using the 45+ specimens of Tylosaurus that the author has personally observed. Do these ontograms match the ones the author augmented with images from other sources?

Validity of the findings

There is certainly validity and importance to reconstructing an ontogenetic series for well-represented mosasaur species, and the effort to do so is clear. Unfortunately, I think the manuscript overreaches from there. The core data are the specimens the author has personally observed and scored into their ontogenetic, cladistic analysis. These should be the focus of the results and discussions sections. There is too much discussion of anagenetic evolution and broad-stroke conclusions about mosasaur ontogeny. These are not only unnecessary, but misleading, because the data are too limited to make many of these claims (I have pointed these out in the PDF).

I think a major refocus of the paper could greatly improve these shortcomings.

Additional comments

I have included my major comments here, but please see the highlighted sections (with comments) in the attached PDF for more details.

- Introductory figures: Figs. 2–5 are hypothetical scenarios that are already summarized in the text. We do not see the first image of a mosasaur bone until Figure 6. If this study is to provide a new character list for assessing ontogeny in tylosaurine mosasaurs, the bulk of the figures need to be devoted to illustrating this anatomical variation. After all, that is the data upon which all of this has to be based. This should not be tucked away in supplementary information.

- Tethysaurus and Mosasaurus ontogeny: given that the author has not personally seen any of this material, this section needs to be removed. There is no primary data collection here and it therefore is entirely based on illustrations and images from other sources. Besides, this detracts from the bulk of the paper, which is actually about Tylosaurus. The paper needs to be re-focused there.

- Discrete growth stage assignment: I have an issue with how growth stages were assigned here. How can a discrete growth stage be assigned without any unambiguous character states to define it? Does this not become an “absence” problem, particularly at the base of an ontogram? Are these discrete states defined by the fact that they don’t show any of the features of the subsequent stages? Moreover, how were all of the continuous variables (e.g. quadrate height, proportion characters) partitioned? Is there good statistical reasons to separate them into their bins, or was it arbitrary? I would highly recommend following the suggestions of Simoes et al. (2016): “Giant taxon-character matrices: quality of character constructions remains critical regardless of size” (Cladistics). They provide explicit guidelines for partitioning these kinds of continuous variables into character states.

- We need to see the anatomical features in the main text for these ontogenetic stages. I am not a fan of relegating all of this important data to the supplementary information. It IS the information upon which all of this relies. The skull/bone images in Figures 6 and 7 are a nice asthetic, but we need to see what the discrete anatomical variation looks like and it needs to be clearly illustrated and labeled.

- Branch support and robustness of the ontograms: these are very poorly supported topologies upon which the conclusions are drawn. This is likely a product of the low sample size and number of characters, but it is important to remember how tenuous such conclusions are from these types of cladograms. I would focus the text on "nodes" or regions of the cladogram for which you have the most data/specimens.

- Postcranial data: this paper is strictly about cranial ontogeny and makes some very bold and broad statements regarding tylosaurine taxonomy and evolution. However, tylosaurines are quite well represented by postcranial data. How might this affect the interpretations made here? It’s a worthwhile thing to discuss, because all too often in mosasaur phylogenetic analyses/taxonomy, the skull (and particularly the quadrate) take precedence over other bits of mosasaur anatomy. Nowhere is this more obvious than in the attempt to reconstruct the ontogram for Tethysaurus: it is nearly entirely partitioned by the quadrate.

- Assignment of “juvenile”, “subadult”, and “adult” statuses: in palaeontology, these are often completely arbitrary and relative terms. I do not know where along the authors’ ontograms I would place these labels, but I take issue with their use of these terms throughout the discussion, particularly when trying to re-diagnose several well-studied taxa. Please see my specific comments in the attached PDF for more detail. These modified cladograms reveal relative “relationships” of individual specimens to each other, but do not reflect the actual ontogenetic status of each individual included in the analysis. These are relative to each other and to an apparently “imaginary adult” and “embryo”. There are severe limitations in terms of what can be said from these analyses and it’s important not to overstretch those boundaries in the discussion.

- Evolutionary and taxonomic implications: as it is currently written, the discussion section really seems to overreach. These ontograms are not really testing for anagenetic evolution or sexual dimorphism. Those data have to come from other sources. In fact, the ontograms contain absolutely zero evolutionary information without a phylogeny. Testing for anagenesis would involve detailed accounts and inclusion of georgraphic, geologic, and phylogenetic information. I would refocus this on the implications for assigning relative ontogenetic status of individual Tylosaurus specimens. There is no reason to go beyond that, given the data presented in the MS.

·

Basic reporting

To whom it may concern:

The study presents, first and foremost, a novel hypothesis concerning possible correlations between certain osteological characters and ontogeny in mosasaurs. This is the very first such an attempt to my knowledge for the taxon in question. The written English is clear and articulate throughout the manuscript, both in the main text and figure/table captions--it was very easy to follow, and I suggested very minor editorial corrections in the annotated PDF. The literature references are both adequate and thorough, and I was unable to make any additional suggestions for more references for the scope of this study.

As far as the article structure/organization was concerned, I believe that Introduction can be a little better organized to deliver the information in a little more logical manner. Specifically, under "Mosasaur ontogeny," the author appears to review both general and genus/species-specific studies on mosasaur ontogeny, yet part of that is found under the subsequent "Tylosaurus proriger" section and onward. I also found Introduction to be rather extensive, unless it's normal for PeerJ. Similarly, the author may consider eliminating Conclusions for the bulk of the contents are already mentioned in the preceding Discussions section, which happens to be also quite extensive and detailed.

As for figures, I find it more helpful that the author include a few representative specimen photos for both Tethysaurus nopcsai and Mosasaurus hoffmannii in their respective ontograms.

Finally, all relevant results of this work have been presented to address new hypotheses.

Experimental design

No comment, except I still find the process of determining ontogenetic stages using osteological characters seemingly circular. The author stated that such osteological characters are independent of absolute size of these mosasaur specimens, in which such characters were either confirmed or absent. At the same time, the author seems to present the logic that osteological traits deemed mature are generally found in larger specimens, and vice versa, which to me appears to introduce a potential circular argument. I don't believe that the proposed ontogenetic markers or traits are entirely independent of the sizes. See my comments on the annotated PDF file, line 288.

Validity of the findings

No comment, but see the above.

Additional comments

I look forward to the publication and continued refinement of this work into the future. I hope my comments will be of benefit.

---

## Round 0.2 · Minor Revisions

Thank you for your careful consideration of all reviewer suggestions. Our reviewer #3 suggests further modifications that will improve your study. I would like you to stick to your data, avoiding interpretations that they do not allow. Please, revise your Figure 14 and your arguments on paedomorphosis related to the parietal table in T. proriger. Take all comments of our reviewer in full consideration. We are almost done.

·

Basic reporting

The changes made improved the reporting style.

Experimental design

I believe the author addresses all of my earlier concerns.

Validity of the findings

I still see no issues with the findings. Limiting the scope of this paper was a prudent decision that strengthens the overall argument.

Additional comments

You were able to satisfy all of my concerns without sacrificing any of your major points in the paper. I believe it is now acceptable for publication in PeerJ.

·

Basic reporting

The basic reporting is clear, contains well-written hypotheses, predictions, and conclusions. Much improved over the first version, particularly after removing Mosasaurus and Tethysaurus from the manuscript. It feels much more focused.

In my opinion, adding figures of several of the osteological features in question to the main text makes it much easier for the reader to follow, rather than switching over to the supplementary information.

Experimental design

The experimental design is in line with other ontogenetic studies done by Carr and others. The methods are very clear now as well.

Validity of the findings

I was concerned that the discussion/conclusions of the first draft overreached a bit, particularly when addressing broader mosasaur ontogeny and evolution. This version is restricted to the core data from the first draft. As a result the findings and their implications are clearly stated and in line with the results from the author's cladistic analyses.

Additional comments

First, let me commend the author on taking the time to consider each of my comments (and those of the other reviewers). I was particularly happy to see the author explore the possibilities of how to partition their continuous variables into discrete characters for their cladistic analysis in the rebuttal letter.

Second, I think removing the sections on Mosasaurus and Tethysaurus makes the manuscript more focused. It flows and follows a single set of research objectives. I do not have any major content-related comments/edits to recommend for this draft.

I have only minor (mostly grammatical, I believe) corrections to the revised draft. I hope you find them helpful.

Sincerely,

Aaron LeBlanc

·

Basic reporting

no comment

Experimental design

no comment

Validity of the findings

Please re-evaluate your argument for the paedomorphosis.

Additional comments

Dear Author,

I have tightened and provided editorial comments to the main text. Introduction and Results sections need least changes, and that is true with the Discussion section up to line 762. After this line, your Discussion section begins to steer away from the original data of your work, which have been well presented and discussed up to this point. Namely, there appear to be an extensive range of future work suggestions accompanied by review of the literature, that seem to stretch out and become defuse. In particular, I strongly suggest that the "Synthesis of ontogeny and phylogeny" section be taken out. I get your points, but this study doesn't answer these issues directly.

Aside from the aforementioned editorial suggestion, which I hope you kindly consider, please consider the following. First, please revise Figure 14 and eliminate any misunderstanding of what constitutes the basioccipital, which is distinct from the pair of exoccipitals, something you seem to consider part of the basioccipital. This is critical because the shape of the occipital condyle plays prominently in shaping your ontogram(s).

Second, I believe the justification for paedomorphosis as the mechanism behind the curved lateral borders of the parietal table in T. proriger may require refinement. Please refer to my comments in the relevant section.

Once again, please consider streamlining your Discussion, as well as addressing the two specific points I raised above--Fig. 14 and your paedomorphosis argument. Once these are addressed and corrected or refined, I believe this important and novel work concerning mosasauroid ontogeny is ready for publication, which I look forward to very much. Last but not least, I thank you very much for your patience waiting for my review.

Sincerely yours,
Takuya Konishi

---

## Round 0.3 · accepted · Accept

I saw that you decided not to follow the suggestions of our reviewer #3 and mine about shortened the discussion section of your paper. Although I do not find your explanation convincing, I do not feel comfortable asking you to delete them, especially considering that two reviewers did not address any problems with this aspect of your work.